taxonomy and systematics

alien species, Caroline Islands, eradication, Mariana Islands, trans-marine dispersal

**Author for correspondence:**
Valter Weijola
e-mail: vsawei@utu.fi

# Taxonomy of Micronesian monitors (Reptilia: Squamata: *Varanus*): endemic status of new species argues for caution in pursuing eradication plans

Valter Weijola[1], Varpu Vahtera[1], André Koch[2],
Andreas Schmitz[3] and Fred Kraus[4]

[1]Zoological Museum, Biodiversity Unit, FI-20014 University of Turku, Turku, Finland
[2]Zoological Research Museum Alexander Koenig, Adenauerallee 160, DE-53113 Bonn, Germany
[3]Natural History Museum of Geneva, Department of Herpetology and Ichthyology, C.P. 6434, CH-1211 Geneva 6, Switzerland
[4]Department of Ecology and Evolutionary Biology, University of Michigan, Ann Arbor, MI 48109, USA

VW, 0000-0002-6907-0619; VV, 0000-0002-6710-6358;
AK, 0000-0003-2426-1494; FK, 0000-0003-4194-4959

In the light of recent phylogenetic studies, we re-assess the taxonomy and biogeography of the *Varanus* populations distributed in the Micronesian islands of Palau, the Western Carolines and the Marianas. Whether these populations are of natural origin or human introductions has long been contentious, but no study has fully resolved that question. Here, we present molecular and morphological evidence that monitor lizards of the *Varanus indicus* Group reached both Palau and the Mariana Islands sometime in the late Pleistocene and subsequently differentiated into two separate species endemic to each geographical region. One species is confined to the Mariana Islands, and for these populations, we revalidate the name *V. tsukamotoi* Kishida, 1929. The other species has a disjunct distribution in Palau, the Western Carolines and Sarigan Island in the Northern Marianas and is herein described as *V. bennetti* sp. nov. Both species are most closely allied to each other, *V. lirungensis* and *V. rainerguentheri*, suggesting that colonization of Micronesia took place from the Moluccas. We discuss the biogeographic distributions of both species in the light of the likely colonization mechanism and previous arguments for human introduction, and we argue that bounties for Palauan populations are ill-advised and plans for eradication of some other populations must first demonstrate that they are, in fact, introduced and not native.

# 1. Introduction

Micronesia is part of one of the world's recognized hotspots of biodiversity (e.g. [1]), though it is depauperate in reptiles compared to many other hotspots. This no doubt reflects the small sum of its land area and its scattered distribution across a vast portion of the Pacific Ocean remote from mainland source areas. Nonetheless, at least 27 endemic reptiles are known from this region, although several remain to be described [2]. Most of the many low-elevation atolls of Micronesia have no or very few endemic species, which instead are concentrated on higher or older islands, especially in Palau [2,3], which is that part of Micronesia most geographically proximate to Asia and New Guinea and, hence, most liable to colonization by dispersing reptiles. Despite this, several endemic Micronesian reptiles are indeed found east of Palau in more-remote Micronesia [4,5] and present interesting biogeographic puzzles regarding their colonization of remote oceanic islands.

Nine species of monitor lizards occur in the *Varanus indicus* Daudin Group (one of three species groups in the subgenus *Euprepiosaurus* Fitzinger), which is widely distributed in the Moluccas, western Melanesia and parts of northern Australia. The northern limits of the *Varanus indicus* Group are Palau, the Carolines and the Mariana Islands, and the presence of these lizards in that region has been reported in the literature since the early 1800s (e.g. [6–8]). Many scientists have regarded these remote populations as human introductions, either by aboriginal populations [9], the German colonial administration prior to WWI [10], or later by the Japanese colonial administration [10–13]. However, based on linguistic evidence and nineteenth-century literature records, Cota [14] argued for either natural colonizations or human-mediated introductions by early Micronesian settlers, at least for populations in the Mariana Islands.

Crombie & Pregill [3] commented on the morphological dissimilarity between populations from Palau and the Mariana Islands and suggested that the monitor lizards of Palau and the Western Carolines (Yap, Ulithi) may be native but the ones in the Eastern Carolines (Ifalik, Kosrae, Pohnpei) and Mariana Islands were introduced. Subsequently, however, fossils that were identified as belonging to the *Varanus indicus* Group were discovered on Guam, thus confirming the presence of the genus *Varanus* well into prehistoric times and rejecting their origin as a recent introduction at least on that island [15], which is the largest of the Mariana Islands. In sum, the origins of many of these populations remain unresolved, though at least some populations seem clearly introduced by humans (Uchida [10]; Buden & Taboroši [5]). Because of this belief in the human origins of many Micronesian populations, removal programmes against these species have been implemented or discussed in several Micronesian jurisdictions (e.g. [16–18], J. Miles 2020, personal communication; D. Vice 2015–2020, personal communication).

Taxonomically, the monitors of Palau, the Western Carolines and the Mariana Islands have all been referred to as *V. indicus* (Daudin, 1802), following Mertens [19]. Earlier, however, Kishida [20]—a Japanese arachnologist and entomologist [21]—had described *Varanus tsukamotoi* from Saipan in a rather obscure natural-history journal [22]. Subsequently, the name *V. tsukamotoi* was synonymized with *V. indicus* by Mertens [19] on the basis of its similarity to that species and Kishida's [20] failure to differentiate his new species from it.

Recently Weijola *et al.* [23] provided a molecular phylogeny for the *Varanus* subgenus *Euprepiosaurus*. This included the *Varanus indicus* Group of that subgenus and showed that the populations from Palau and the Mariana Islands are genetically distinct both from each other and from all other known species in the group. Their findings also showed that all these assorted populations assigned to or recently partitioned from *V. indicus* form a monophyletic clade that diverged from each other sometime during the late Pleistocene (*ca* 1–1.2 Ma), well prior to human presence in this insular region. This supported the biogeographic hypothesis that monitor lizards are indeed native to both Micronesian island groups rather than recent human introductions, although this does not refute that humans have artificially expanded their distributions by introducing them to additional islands [5,10,11,24]. This is of especial importance in the Federated States of Micronesia, where many islands have introduced populations, other islands have populations of uncertain status (i.e. native versus introduced) [5], and for which Weijola *et al.* [23] lacked samples.

Herein, we add additional species and populations to the sampling of Weijola *et al.* [23], present additional new molecular sequences, and add morphological data to assess the evolutionary distinctiveness of the Palauan and Marianan monitor lizards. These include newly added Moluccan taxa that are critical for understanding the evolution and biogeography of the Micronesian monitors but that were absent in Weijola *et al.* [23]. We reinstate the name *Varanus tsukamotoi* Kishida, 1929—which has been placed in synonymy for more than 75 years—for those populations inhabiting

**Table 1.** Definitions of, and abbreviations used for, measurements, proportion indices and scale counts.

| symbol | description |
| --- | --- |
| *measurements* | |
| SVL | snout-to-vent length |
| F | tail length |
| TL | total length |
| A | head length from snout to anterior dorsal margin of tympanum |
| B | head width at maximum span of postorbital arch |
| C | head depth at midpoint of orbit |
| G | facial length from centre of nostril to anterior margin of orbit |
| H | snout length from tip of snout to centre of nostril |
| *proportion indices* | |
| 1 | relative tail-to-body length—F/SVL |
| 10 | relative head length to width—A/B |
| 11 | relative head length to height—A/C |
| *scale counts* | |
| S | midbody scale rows |
| XY | dorsal scale rows from dorsal margin of tympanic recess to anterior margin of hind limbs |
| T | transverse rows of ventral scales from gular fold to anterior margin of hind limbs |
| X | transverse rows of dorsal scales from posterior margin of tympanic recess to gular fold |
| m | scales around neck at anterior margin of gular fold |
| N | rows of ventral scales from tip of snout to gular fold |
| P | scales from rictus to rictus across dorsum of head |
| Q | scales around tail base |
| R | scales around tail counted at one-third of the length from the base |
| DOR | number of dorsal scale rows from the last occipital scale to a point dorsal to the posterior margin of the cloaca |
| VEN | ventral scales from the gular fold to the anterior margin of the cloaca |

the Mariana Islands (except Sarigan), and we describe the Palauan, Western Caroline and Sarigan populations as a new species. This increases the number of known endemic reptile species in Micronesia, demonstrates the exceptional ability for overseas dispersal of this group of lizards and argues for careful evaluation of the origin of each island's monitor populations prior to considering eradication or control programmes.

# 2. Material and methods

## 2.1. Morphology

We scored new scalational and meristic data of the two candidate species from museum specimens. Definitions of, and abbreviations used for, measurements, proportion indices and scale counts follow Brandenburg [25] and Weijola & Sweet [26] (table 1). We took measurements to the nearest 0.5 mm (head) or 1 mm (SVL, tail) with a steel tape or callipers. Comparative scalational and meristic data for *V. cerambonensis*, *V. douarrha*, *V. indicus*, *V. juxtindicus*, *V. lirungensis*, *V. melinus*, *V. obor*, *V. rainerguentheri* and *V. zugorum* were obtained from the literature [25–32].

We conducted *a posteriori* linear discriminant analysis in PAST v. 3.24 [33] on scalational characters P, Q, XY, m, S, T, N and R, including candidate species 1 from the Marianas ($n = 26$) and Japtan Island (Marshall Islands) ($n = 3$); candidate species 2 from Palau ($n = 8$), the Western Carolines ($n = 7$) and

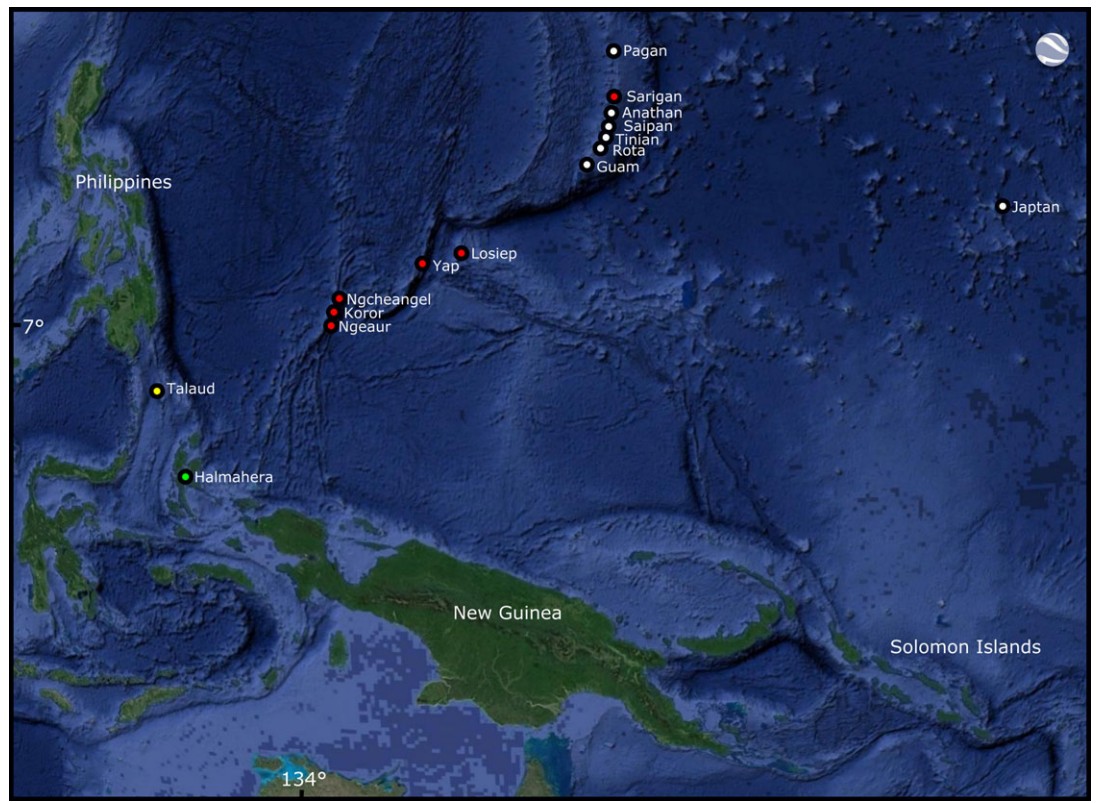

**Figure 1.** Map of the Pacific region showing the distribution of *Varanus tsukamotoi* (white dots), *V. bennetti* sp. nov. (red dots), *V. lirungensis* (yellow dot) and *V. rainerguentheri* (green dot).

Sarigan ($n = 1$); and their two closest relatives *V. lirungensis* ($n = 9$) from the Talaud Islands and *V. rainerguentheri* ($n = 7$) from Halmahera Island in the Moluccas, in order to evaluate how well the species and candidate species within this clade can be discriminated in multivariate space. We visualized body proportion index 1 (F/SVL) for the same specimens with a box-plot graph in order to show intra- and interspecific variation and differences for that character (data used in the morphological analyses can be found in electronic supplementary material, file 1).

## 2.2. Museum abbreviations used

Australian Biological Tissue Collection, Adelaide (ABTC); Australian Museum, Sydney (AMS); Bernice Pauahi Bishop Museum, Honolulu (BPBM); California Academy of Sciences, San Francisco (CAS); Forschungsinstitut und Natur-Museum Senckenberg, Frankfurt (SMF); Museum Zoologicum Bogoriense, Bogor (MZB), Natural History Museum, London (BMNH) Queensland Museum, Brisbane (QM); United States National Museum, Washington, DC (USNM); University of Michigan Museum of Zoology, Ann Arbor (UMMZ); Western Australian Museum, Perth (WAM); Zoologisches Forschungsmuseum Alexander Koenig, Bonn (ZFMK); Zoological Museum Hamburg, Hamburg (ZMH) and the Zoological Museum of the University of Turku (ZMUT).

## 2.3. Molecular genetics

### 2.3.1. Taxon sampling

Our dataset contains 51 samples divided among 11 recognized and eight candidate species (including those described below) of the subgenus *Euprepiosaurus* Fitzinger, 1843. We sequenced a total of 12 new samples of six species: *V. cerambonensis* (Seram Island), *V. indicus* (Karkar Island, Sakar Island, Tolokiwa Island, Umboi Island), *V. lirungensis* (Talaud Islands), *V. rainerguentheri* (Halmahera Island) and the two candidate species from Micronesia (Ngeaur Island, Sarigan Island, Saipan Island) (figure 1), and we added these to the taxa sampled in Weijola *et al*. [23]. We included *Varanus reisingeri* from the sister group of *Euprepiosaurus* as outgroup (table 2).

**Table 2.** Specimens included in the molecular phylogenetic studies and GenBank numbers for ND4 and 16S sequences. New sequences shown in bold.

| species | voucher | locality | country | ND4/16S GenBank acc. no. |
|---|---|---|---|---|
| *Varanus bennetti* sp. nov. | CAS238226 | Ngcheangel Atoll | Palau | MK388753/MK388747 |
| *Varanus bennetti* sp. nov. | USNM514125 | Ngeaur Island | Palau | MK388754/MK388748 |
| *Varanus bennetti* sp. nov. | USNM536546 | Sarigan, Mariana Islands | CNMI | **MT011991/MN978730** |
| *Varanus bennetti* sp. nov. | USNM507504 | Ngeaur | Palau | **MT011990/MN978731** |
| *V. cerambonensis* | WAM109448 | Banda, Moluccas | Indonesia | KU513445/KU513465 |
| *V. cerambonensis* | WAM109476 | Banda, Moluccas | Indonesia | KU513446/KU513466 |
| *V. cerambonensis* | ZFMK70618 | Seram, Moluccas | Indonesia | **MT011987**/EF193653 |
| *V. doreanus* | UMMZ227117 | Merauke, West Papua | Indonesia | KU513448/KU513468 |
| *V. doreanus* | BPBM19509 | Mt Obree, Northern Province | PNG | KU513447/KU513467 |
| *V. douarrha* | ZMUT Sa159 | Fissoa, New Ireland | PNG | KY770830/KY770804 |
| *V. douarrha* | ZMUT Sa173 | Bangun, Lavongai Island | PNG | KY770836/KY770810 |
| *V. douarrha* | ZMUT Sa174 | Djaul Island | PNG | KY770835/KY770809 |
| *V. finschi* | ZMUT Sa186 | Nodup, East New Britain | PNG | KU513443/KU513463 |
| *V. finschi* | ZMUT Sa190 | Kokopo, East New Britain | PNG | KU513444/KU513464 |
| *V. indicus* | ZMUT Sa182 | Duke of York Island | PNG | KY770838/KY770812 |
| *V. indicus* | ZMUT Sa188 | Watom Island, East New Britain | PNG | KY770833/KY770807 |
| *V. indicus* | ZMUT Sa202 | Koki, East New Britain | PNG | KU513456/KU513477 |
| *V. indicus* | LSUMZ H10449 | Wewak, East Sepik | PNG | KU513451/KU513472 |
| *V. indicus* | ABTC99075 | Wegamu, Trans Fly | PNG | KY770839/KY770813 |
| *V. indicus* | AMSR137997 | Fergusson Island, Milne Bay | PNG | KU513450/KU513471 |
| *V. indicus* | WAM109764 | Kai Besar, Moluccas | Indonesia | KY770847/KY770821 |
| *V. indicus* | WAM109525 | Aru Islands, Moluccas | Indonesia | KU513453/KU513474 |
| *V. indicus* | UMMZ248339 | Umboi | PNG | **MT011995**/– |

**Table 2.** (*Continued.*)

| species | voucher | locality | country | ND4/16S GenBank acc. no. |
|---|---|---|---|---|
| *V. indicus* | UMMZ248338 | Sakar | PNG | **MT011996**/– |
| *V. indicus* | UMMZ248329 | Tolokiwa | PNG | **MT011994**/– |
| *V. indicus* | UMMZ248328 | Karkar | PNG | **MT011997**/– |
| *V. jobiensis* | AMSR116999 | Wigote, Torricelli Mts. West Sepik | PNG | KU513457/KU513479 |
| *V. jobiensis* | AMSR115341 | Doido, Chimbu Province | PNG | DQ525163/KU513478 |
| *V. lirungensis* | MZB_Lac5178 | Talaud | Indonesia | **MT011988**/EF193670 |
| *V. lirungensis* | MZB_Lac5177 | Talaud | Indonesia | **MT011989**/EF193671 |
| *V. melinus* | UMMZ222682 | Sula Islands, Moluccas | Indonesia | KU513458/KU513480 |
| *V. rainerguentheri* | ZFMK85404 | Halmahera, Moluccas | Indonesia | **MT011998**/EF193659 |
| *V. semotus* | ZMUT Sa176 | Mussau Island, St Matthias group | PNG | KU513459/KU513482 |
| *V. semotus* | ZMUT Sa177 | Mussau Island, St Matthias group | PNG | KU513460/KU513483 |
| *Varanus* sp. | ZMUT Sa169 | Los Negros, Admiralty Islands | PNG | MK388782/MK388738 |
| *Varanus* sp. | ZMUT Sa171 | Kawaliap, Manus Island | PNG | MK388780/MK388735 |
| *Varanus* sp. | AMSR121569 | Shortland Island | Solomon Islands | KY770843/KY770817 |
| *Varanus* sp. | AMSR134949 | New Georgia | Solomon Islands | KY770844/KY770818 |
| *Varanus* sp. | AMSR134950 | New Georgia | Solomon Islands | KY770845/KY770819 |
| *Varanus* sp. | WAM109969 | Tanimbar, Moluccas | Indonesia | MK388749/MK388745 |
| *Varanus* sp. | WAM112255 | Tanimbar, Moluccas | Indonesia | MK388750/MK388746 |
| *Varanus* sp. | ZMUT Sa201 | Rossell Isl., Louisiades | PNG | MK388775/MK388732 |
| *Varanus* sp. | ZMUT Sa 197 | Misima Isl., Louisiades | PNG | MK388776/MK388727 |
| *Varanus* sp. | ZMUT Sa200 | Sudest Isl., Louisiades | PNG | MK388778/MK388733 |
| *Varanus* sp. | ABTC76130 | Barova Faa Island | Solomon Islands | MK388786/– |
| *Varanus* sp. | ABTC76131 | Kolopakisa, Isabel Island | Solomon Islands | MK388787/– |

(*Continued.*)

**Table 2.** (*Continued*.)

| species | voucher | locality | country | ND4/16S GenBank acc. no. |
|---|---|---|---|---|
| *V. tsukamotoi* | UMMZ241997 | Guam, Mariana Islands | USA | MK388751/MK388739 |
| *V. tsukamotoi* | UMMZ241999 | Guam, Mariana Islands | USA | MK388752/MK388740 |
| *V. tsukamotoi* | USNM576259 | Saipan, Mariana Islands | CNMI | **MT011993/MN978732** |
| *V. tsukamotoi* | USNM576257 | Saipan, Mariana Islands | CNMI | **MT011992/MN978733** |
| *V. yuwonoi* | UMMZ225545 | Halmahera, Moluccas | Indonesia | KU513462/KU513481 |
| Outgroup | | | | |
| *V. reisingeri* | No Voucher | Pet trade (Misool, Raja Ampat?) | Indonesia | KY770849/KY770823 |

## 2.3.2. Laboratory methods

We sequenced a segment of the mitochondrial protein-coding ND4 and mitochondrial ribosomal 16S rDNA in order to produce a dataset consistent with previous studies [23,30]. We extracted total DNA from liver tissue fixed in 96% EtOH using either the NucleoSpin® Tissue Kit (Macherey-Nagel) following the standard protocol for animal tissue or QIAamp DNA Mini Kit (Qiagen). ND4 was amplified using the forward primer 5′ TGA CTA CCA AAA GCT CAT GTA GAA GC 3′ [34] and the reverse primer 5′ CAT TAC TTT TTA CTT GGA TTT GCA CCA 3′ [35]. The PCR amplification mixture for ND4 consisted of 2 µl template DNA, 7.5 µl MQ water, 12.5 µl of MyTaq™ Red Mix (Bioline) and 0.5 µl of forward and reverse primers. The amplification cycle consisted of an initial denaturation of 1 min at 95°C, followed by 35 cycles of denaturation of 15 s at 95°C, annealing of 15 s at 46.5°C, followed by extension of 10 s at 72°C. 16S rDNA was amplified using the primer pair 5′-CGC CTG TTT ATC AAA AAC AT- 3′ (forward) / 5′ -CCG GTC TGA ACT CAG ATC ACG T- 3′ (reverse) [36]. The amplification was performed with Phusion U Hot Start PCR Master Mix (ThermoFisher Scientific, St Leon-Rot, Germany) following the manufacturer's protocol. The amplification profile consisted of an initial denaturation of 9 min at 94°C, followed by 35 cycles of denaturation of 30 s at 94°C, annealing of 25 s at 55°C, followed by extension of 35 s at 72°C and 2 min final elongation step at 72°C. A negative control without DNA was included in every amplification run. PCR products were visualized by 1% agarose electrophoresis using Midori Green Advanced DNA Stain. Amplified fragments were next purified with the A'SAP PCR cleanup kit (ArcticZymes) and then sent to Macrogen Europe for sequencing. The quality of the chromatograms was inspected with Sequencer 5 (Gene Codes Corporation, USA). All new sequences are deposited in GenBank (table 2).

## 2.4. Phylogenetic analysis

The ND4 dataset contained 52 nucleotide sequences (including outgroup) and 651 bps; the 16S dataset contained 45 sequences and 349 bps. We visualized the sequences and trimmed their ends so that they were all of equal length using Mesquite v. 3.1 [37], after which we concatenated the ND4 and 16S datasets with SequenceMatrix [38]. We calculated raw pairwise distances between species and candidate species in MEGA v. 7 [39].

To conduct parsimony analyses of the combined dataset, we used TNT v. 1.5 [40]. Our search strategy consisted of 100 replications and of 10 rounds of both ratchet and tree drifting followed by tree fusing [41]. We executed the command xmult until 50 independent hits of the shortest tree were found and produced a strict consensus of the most-parsimonious (MP) trees. We also applied the command 'blength' to report the branch lengths of the resulting trees. To estimate nodal support we used the Jackknife [42] resampling method with 1000 replicates and a probability of character removal of 0.36.

For the likelihood analyses, we used RAxML v. 8.2.10 [43] on the CIPRES Science Gateway platform [44]. Since RAxML only implements GTR-based models of nucleotide substitutions, we used the unique general time-reversible model and estimated nodal support with the rapid-bootstrap algorithm (1000 replications).

# 3. Results

## 3.1. Morphology

The linear discriminant analysis based on scalational features assigned 96.7% of samples to the correct species and retrieved a clear group structure for the included named and candidate species. In particular, *Varanus* sp. 1 from the Mariana Islands was clearly different from the other three species compared (figure 2*a*). The only two species to show a partial overlap in multivariate space are *Varanus* sp. 2 from Palau, Western Carolines and Sarigan and *V. rainerguentheri* (figure 2*a*); however, sample size for *V. rainerguentheri* was small, not allowing for confident discrimination of the two species with scalational features alone. Scalational characters P, S and T accounted for most of the differences in our analysis (table 3), and they show virtually no overlap between the two candidate species, with candidate species 1 from the Mariana Islands having significantly lower values than *V. lirungensis*, *V. rainerguentheri* and the candidate species 2 from Palau, Western Carolines and Sarigan (table 4). The most distinguishing morphometric feature for candidate species 2 is its tail

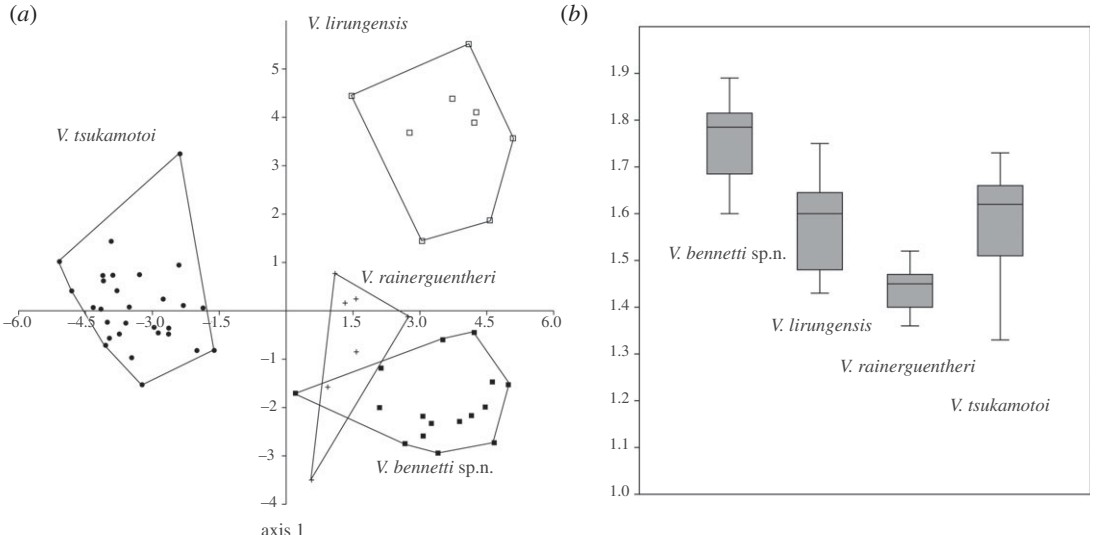

**Figure 2.** (*a*) Linear discriminant function analysis of scalational characters for *V. bennetti* sp. nov., *V. lirungensis*, *V. rainerguentheri* and *V. tsukamotoi* showing non-overlapping multivariate morpho-spaces for all species except *V. bennetti* sp. nov. and *V. rainerguentheri*. (*b*) Box-plot of relative tail length (proportion index 1) for examined individuals of *V. bennetti* sp. nov., *V. lirungensis*, *V. rainerguentheri* and *V. tsukamotoi* showing the exceptionally long tail of *V. bennetti* sp. nov.

**Table 3.** Factor loadings of the linear discriminant analysis, showing P, S and T to be the meristic characters providing greatest discrimination among the species.

| factor | Axis 1 | Axis 2 | Axis 3 |
|---|---|---|---|
| P | 0.155 | −0.003 | −0.286 |
| Q | 0.019 | −0.042 | 0.028 |
| XY | −0.007 | −0.106 | 0.009 |
| m | 0.000 | −0.025 | 0.154 |
| S | 0.119 | 0.123 | −0.040 |
| T | 0.150 | 0.168 | 0.096 |
| N | 0.055 | −0.224 | 0.099 |
| R | −0.009 | −0.062 | 0.075 |
| eigenvalue | 11.39 | 3.22 | 0.29 |
| per cent | 76.44 | 21.63 | 1.93 |

length (F), which is proportionately the longest of any species in the *V. indicus* group at 1.6–1.89 (average 1.76) times SVL and clearly distinguishes it from *V. rainerguentheri* (figure 2*b* and table 5), the only species it overlapped with in the discriminant-function analysis.

## 3.2. Molecular phylogenetics

Parsimony analysis resulted in nine MP trees of length 572 steps, of which the strict consensus tree has *Varanus* sp. from Palau and Sarigan grouping together as a strongly supported clade, with a jackknife value of 99 (hereafter JF) and six synapomorphies (hereafter syn) (figure 3). This species further groups together with *Varanus* sp. 1 from the Marianas and *V. lirungensis*, and these three species form a strongly supported (JF 96, syn 16), monophyletic group. The nine MP trees disagree on the internal relationships among these three taxa and are therefore left unresolved in the strict consensus tree. Likewise, the nine best trees disagree on the evolutionary placement of this clade within the *V. indicus* group. Despite the strict consensus tree leaving many of the internal relationships within the

**Table 4.** Ranges and averages for scale count characters of species within the *V. indicus* Group.

| | P | Q | R | S | T | N | X | XY | M |
|---|---|---|---|---|---|---|---|---|---|
| *V. bennetti* sp. nov. (*n* = 16) | 39–49 (43.3) | 74–96 (87.1) | 55–68 (62.1) | 131–145 (136.9) | 92–100 (95.7) | 87–98 (91.8) | 41–48 (45.9) | 148–160 (153.8) | 85–117 (101.6) |
| *V. cerambonensis* Ambon, Seram (*n* = 11) | 46–54 (50.1) | 79–91 (85.5) | 49–60 (53.5) | 131–150 (141.4) | 93–100 (96.3) | 79–91 (85.5) | 35–43 (39.9) | 138–154 (147.7) | 96–110 (103.3) |
| *V. douarrha* (*n* = 14) | 39–50 (43.5) | 83–104 (90.2) | 49–64 (57.5) | 129–153 (140.9) | 90–108 (96.9) | 82–92 (85.7) | 35–44 (39.9) | 125–147 (136) | 91–115 (100.7) |
| *V. indicus* New Guinea (*n* = 40) | 36–47 (42) | 60–99 (78.4) | 46–69 (60.8) | 100–145 (124) | 74–96 (88) | 71–94 (78.9) | 28–42 (36.4) | 109–158 (127.8) | 73–103 (87.3) |
| *V. juxtindicus* (*n* = 7) | 39–43 (41) | 72–87 (78) | 47–52 (49.5) | 128–140 (135.2) | 104–111 (106.6) | 83–96 (87.6) | 34–43 (38.2) | 132–149 (140.2) | 92–104 (97.4) |
| *V. lirungensis* (*n* = 11) | 38–47 (42.4) | 79–88 (83) | 55–65 (59.5) | 134–151 (141.6) | 92–102 (97.5) | 81–88 (84.1) | 34–48 (39) | 131–159 (139.8) | 87–106 (97.8) |
| *V. melinus* (*n* = 4) | 46–55 (49.3) | 81–85 (83) | 58–68 (63.3) | 124–133 (128.3) | 87–99 (93) | 83–89 (86.5) | 29–45 (37.8) | 119–151 (134.3) | 83–91 (86.8) |
| *V. obor* (*n* = 1/7*) | 47 | 80 | — | 119–134 (125.4*) | 88–94 (92.4*) | — | 35 | 119–127 (123.4*) | 85 |
| *V. rainerguentheri* (*n* = 7) | 38–44 (42.1) | 76–82 (79.1) | 57–61 (58.3) | 120–139 (128.9) | 90–94 (91.9) | 85–92 (87.6) | 38–47 (41.7) | 133–163 (144) | 85–94 (89.4) |
| *V. tsukamotoi* (*n* = 41) | 31–40 (34.8) | 54–74 (67) | 48–59 (54) | 101–126 (113.5) | 78–88 (84) | 74–90 (79.9) | 33–45 (39.2) | 117–140 (131) | 76–103 (86.4) |
| *V. zugorum* (*n* = 1) | 45 | 82 | 56 | 134 | 97 | 101 | 35 | 128 | 105 |

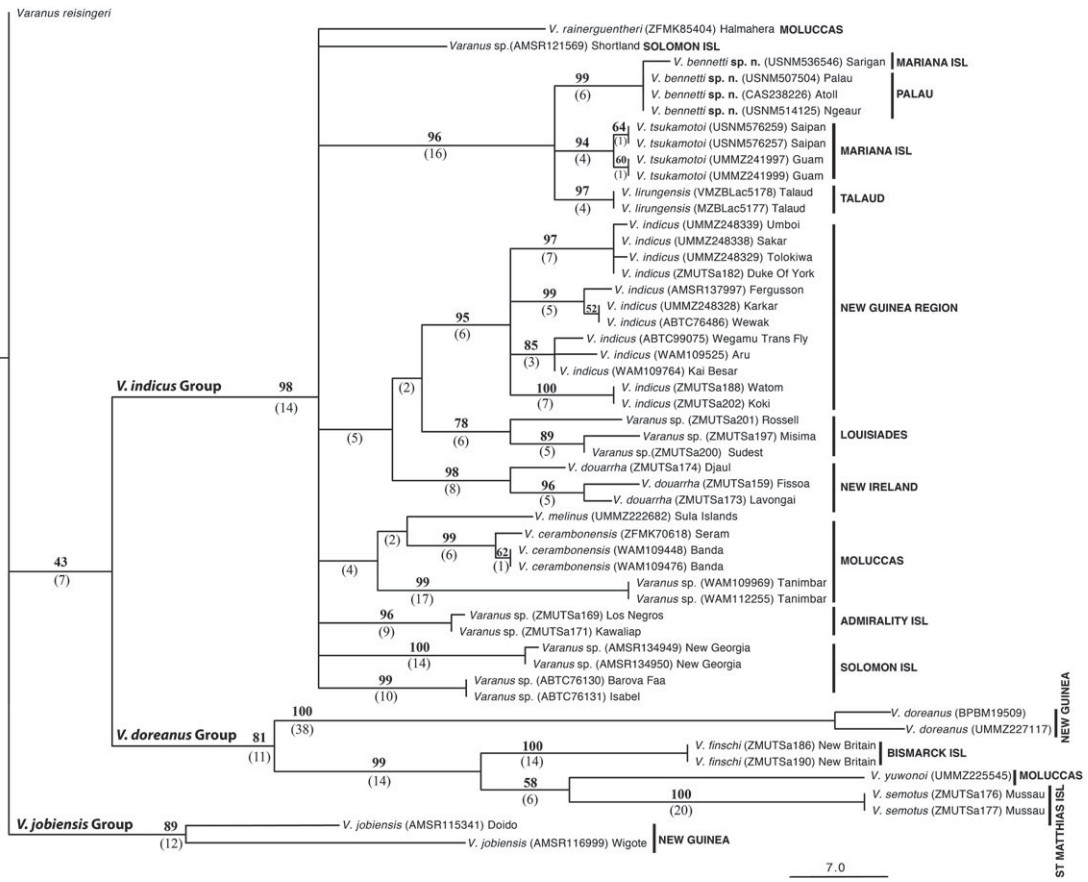

**Figure 3.** Strict consensus of the nine most-parsimonious trees resulting from the TNT analysis. Jackknife resampling values greater than 50% are shown above, and the numbers of synapomorphies shared by each clade are shown below the nodes in parentheses. Branch lengths represent the number of optimized character-state changes.

**Table 5.** Ranges and averages for body proportion indices within the *V. indicus* Group.

|  | 1 (F/SVL) | 10 (A/B) | 11 (A/C) |
|---|---|---|---|
| *V. bennetti* sp. nov. (*n* = 16) | 1.60–1.89 (1.76) | 1.63–2.30 (1.85) | 2.36–2.88 (2.60) |
| *V. cerambonensis* Ambon, Seram (*n* = 11) | 1.32–1.61 (1.50) | 1.77–2.17 (1.95) | 2.50–2.94 (2.76) |
| *V. douarrha* (*n* = 14) | 1.29–1.61 (1.48) | 1.67–2.00 (1.85) | 2.42–2.86 (2.58) |
| *V. indicus* New Guinea (*n* = 40) | 1.22–1.70 (1.49) | 1.68–2.00 (1.86) | 2.44–2.70 (2.59) |
| *V. juxtindicus* (*n* = 7) | 1.50–1.74 (1.58) | 1.37–1.88 (1.71) | 2.14–2.63 (2.46) |
| *V. lirungensis* (*n* = 11) | 1.43–1.75 (1.58) | 1.71–2.18 (1.87) | 2.25–2.66 (2.51) |
| *V. melinus* (*n* = 4) | 1.65–1.79 (1.73) | 2.23–2.39 (2.33) | 2.75–3.06 (2.95) |
| *V. obor* (*n* = 7) | 1.49–1.66 (1.58) | 1.94–2.20 (2.06) | — |
| *V. rainerguentheri* (*n* = 7) | 1.36–1.47 (1.44) | 1.84–2.21 (2.00) | 2.55–3.47 (2.86) |
| *V. tsukamotoi* (*n* = 41) | 1.33–1.73 (1.58) | 1.54–2.0 (1.78) | 2.23–2.87 (2.6) |
| *V. zugorum* (*n* = 1) | 1.40 | 1.97 | 2.69 |

*V. indicus* species group unresolved, all included species are well supported and characterized by several molecular synapomorphies.

Likelihood analysis was congruent with the parsimony analysis in the placement of *Varanus* sp. 2 from Palau and Sarigan (bootstrap value 99, here after BS 99) together with *Varanus* sp. 1 from the Marianas (BS 93) and *V. lirungensis* (BS 100), but, as for the strict parsimony consensus, it did not resolve relationships among the three species (figure 4). However, it also placed *V. rainerguentheri*

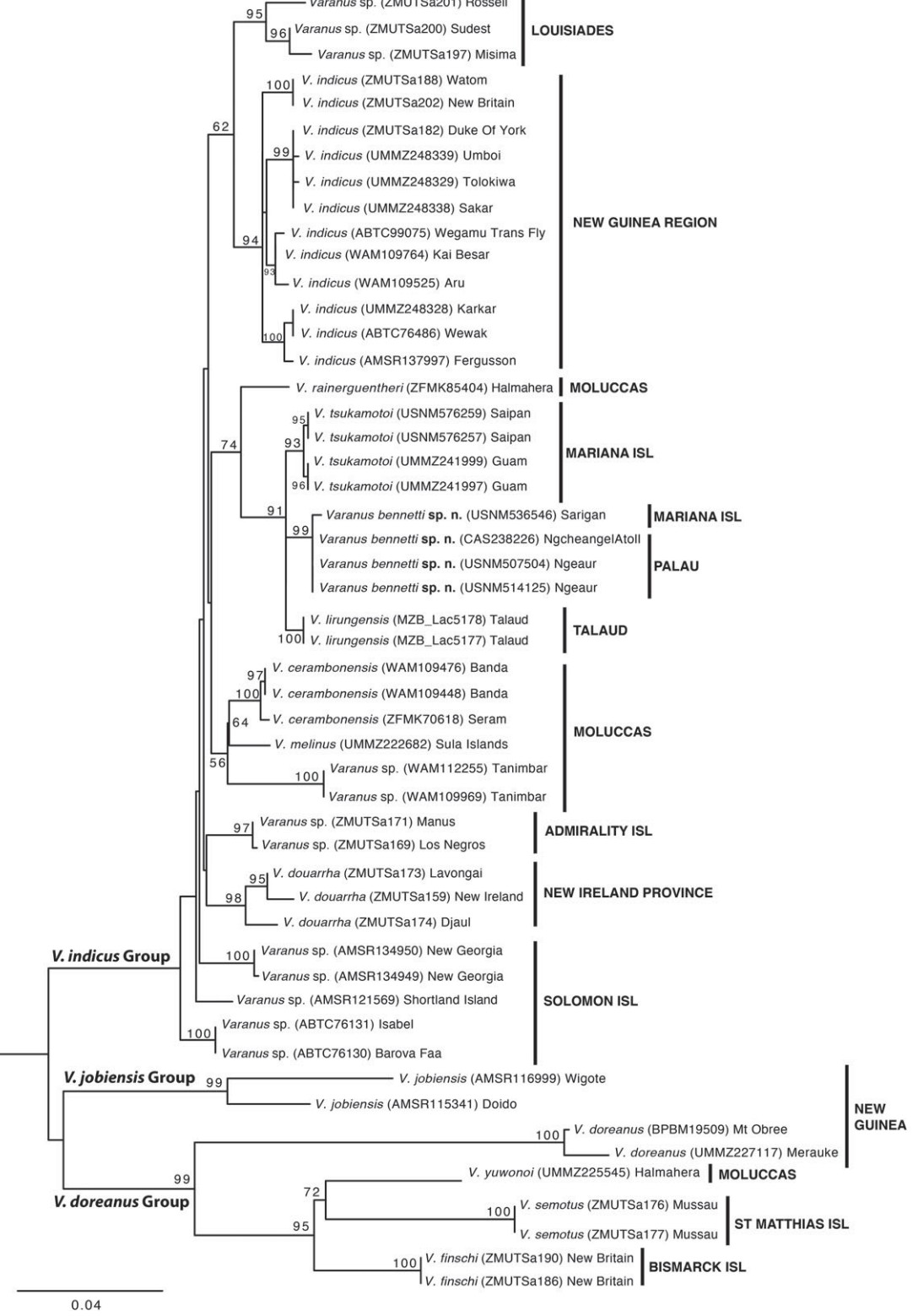

**Figure 4.** Optimal maximum-likelihood tree resulting from the RAxML analysis. Bootstrap support values greater than 50% are shown on the nodes. Scale bar corresponds to the mean number of nucleotide substitutions per site.

basal to this group with moderate (BS 74) nodal support. Also similar to the parsimony results, the internal relationships within the *V. indicus* species group were characterized by low nodal support, whereas individual species had high support. Importantly, although *V. rainerguentheri* and *Varanus*

sp. 2 could not be well differentiated using only scalational features (figure 2*a*), molecular data clearly separate the two (figures 3 and 4).

*Varanus* sp. 2 from Palau differs from *Varanus* sp. 1 from the Marianas by 1.5/0.3% (ND4 and 16S, respectively) in mean distance, from *V. lirungensis* by 1.4/0.3% and from *V. rainerguentheri* by 3.1/0.3% (table 6). For the two candidate taxa, minor intraspecific differences were also found between the populations of Saipan and Guam, and the populations of Ngeaur and Sarigan (figures 3 and 4).

## 3.3. Taxonomy

Our molecular and morphological data both make clear that the Micronesian lineages studied here and historically referred to as *Varanus indicus* are in fact separate phylogenetic lineages that have been geographically and genetically isolated from each other and other members of their species group since the late Pleistocene [23], and both lineages are diagnosable by meristic and morphometric characters. The combination of both lines of evidence suggests that they are independently evolving lineages and they fulfil the definition of species according to the unified species concept applicable to allopatric species [45]. The species from Saipan and surrounding islands was earlier described as *V. tsukamotoi* [20]; however, the original description and diagnosis of *V. tsukamotoi* was vague and insufficient to differentiate the species against the many species currently recognized in the *V. indicus* Group. We take this opportunity to redescribe this poorly diagnosed taxon and to describe the *Varanus* populations occurring in Palau, the Western Carolines and Sarigan Island in the Northern Marianas as a new species. Further, the type specimen of *V. tsukamotoi* has been lost. Given the presence of two species of the *V. indicus* Group in the Marianas Islands, to ensure future taxonomic stability, we hereby designate a neotype for this species.

## 3.4. *Varanus tsukamotoi* Kishida, 1929, figures 5–7

### 3.4.1. Neotype

Although Kishida designated a holotype (a young female collected on Saipan by Inenosuke Iwai), he did not provide a catalogue number, and the specimen in question is now considered lost, as seems to be the case with most of Kishida's collections [46,47]. To promote taxonomic stability, inasmuch as two *Varanus* species of the *V. indicus* Group inhabit the Mariana Islands, we designate USNM 576258 (figure 5*a,b*) as neotype for this species. The specimen was collected by Ronald Crombie on 15 July 1999 in a scrubby forest at Makpi at the north end of Saipan Island in the Commonwealth of the Northern Marianas.

### 3.4.2. Additional specimens examined

Mariana Islands: Anatahan (USNM 212496); Cocos Island (USNM 494383); Guam (AMNH 137204, USNM 121793, 122462–63, 122465, 122468, 121796, 216368, 216869, 323717, 494380–82, 494582, 507452, 515827–29, 521160); Pagan (USNM 212489, 212491–93); Rota (AMNH 137202–03, 139986–88, USNM 122652–53, 285037); Saipan (SMF 30152–53, USNM 212487–88, 576257, 576259, ZMB 22645); Tinian (USNM 308067); indefinite locality (MNHN 1888 20); Marshall Islands: Enewetak Atoll, Japtan Island (AMNH 78994, USNM 124112–13).

### 3.4.3. Diagnosis

*Varanus tsukamotoi* can be distinguished from all other members of the *V. indicus* group by its unique combination of: (i) dorsum black and covered with evenly distributed yellow scales, (ii) tongue dark blue/grey, (iii) yellow temporal stripe usually absent, (iv) low scale counts around the head (P: 31–40), tail base (Q: 54–74) and midbody (S: 101–126), and (v) usually prominent dark pigmentation in the gular region.

### 3.4.4. Comparisons with other members of *Euprepiosaurus*

*Varanus tsukamotoi* can be distinguished from *V. caerulivirens, V. colei, V. doreanus, V. finschi, V. jobiensis, V. juxtindicus, V. melinus, V. obor, V. semotus* and *V. yuwonoi* by having an entirely dark blue/grey tongue rather than (at least partly) pink or yellow tongue; from *V. cerambonensis* by the absence of a yellow temporal stripe, dorsal bands composed of yellow scales and lower midbody scale counts (S: 101–126 versus 131–150 in *V. cerambonensis*); from *V. douarrha* by the absence of dorsal ocelli and lower

**14**

**Table 6.** Pairwise distances of ND4 and 16S sequences for species and candidate species of *Euprepiosaurus* included in the phylogenetic analysis.

| | 1 | 2 | 3 | 4 | 5 | 6 | 7 | 8 | 9 | 10 | 11 | 12 | 13 | 14 | 15 | 16 | 17 | 18 |
|---|---|---|---|---|---|---|---|---|---|---|---|---|---|---|---|---|---|---|
| **ND4** | | | | | | | | | | | | | | | | | | |
| **1** V. doreanus | | | | | | | | | | | | | | | | | | |
| **2** V. yuwonoi | 0.121 | | | | | | | | | | | | | | | | | |
| **3** V. finschi | 0.116 | 0.069 | | | | | | | | | | | | | | | | |
| **4** V. semotus | 0.116 | 0.065 | 0.063 | | | | | | | | | | | | | | | |
| **5** Varanus sp. Tanimbar | 0.130 | 0.117 | 0.118 | 0.108 | | | | | | | | | | | | | | |
| **6** V. tsukamotoi | 0.122 | 0.106 | 0.111 | 0.109 | 0.048 | | | | | | | | | | | | | |
| **7** V. lirungensis | 0.125 | 0.106 | 0.111 | 0.106 | 0.048 | 0.013 | | | | | | | | | | | | |
| **8** V. bennetti **sp. nov.** | 0.124 | 0.111 | 0.113 | 0.105 | 0.050 | 0.015 | 0.014 | | | | | | | | | | | |
| **9** V. rainerguentheri | 0.121 | 0.111 | 0.112 | 0.101 | 0.043 | 0.030 | 0.029 | 0.031 | | | | | | | | | | |
| **10** V. douarrha | 0.127 | 0.131 | 0.124 | 0.112 | 0.053 | 0.052 | 0.050 | 0.049 | 0.042 | | | | | | | | | |
| **11** V. indicus | 0.122 | 0.122 | 0.118 | 0.108 | 0.047 | 0.038 | 0.036 | 0.038 | 0.043 | 0.039 | | | | | | | | |
| **12** Varanus sp. Louisiades | 0.120 | 0.119 | 0.110 | 0.108 | 0.047 | 0.045 | 0.038 | 0.042 | 0.042 | 0.040 | 0.033 | | | | | | | |
| **13** V. cerambonensis | 0.117 | 0.110 | 0.118 | 0.102 | 0.036 | 0.041 | 0.037 | 0.042 | 0.033 | 0.038 | 0.041 | 0.039 | | | | | | |
| **14** V. melinus | 0.125 | 0.126 | 0.131 | 0.111 | 0.040 | 0.045 | 0.045 | 0.043 | 0.034 | 0.042 | 0.043 | 0.041 | 0.024 | | | | | |
| **15** Varanus sp. Admiralties | 0.114 | 0.118 | 0.110 | 0.105 | 0.042 | 0.035 | 0.035 | 0.033 | 0.030 | 0.034 | 0.037 | 0.031 | 0.030 | 0.033 | | | | |
| **16** Varanus sp. Isabel | 0.119 | 0.117 | 0.112 | 0.101 | 0.034 | 0.039 | 0.035 | 0.037 | 0.031 | 0.033 | 0.032 | 0.034 | 0.033 | 0.037 | 0.030 | | | |
| **17** Varanus sp. Shortland I. | 0.118 | 0.115 | 0.111 | 0.097 | 0.037 | 0.042 | 0.041 | 0.043 | 0.028 | 0.033 | 0.040 | 0.040 | 0.027 | 0.031 | 0.027 | 0.025 | | |
| **18** Varanus sp. New Georgia | 0.124 | 0.122 | 0.114 | 0.100 | 0.036 | 0.045 | 0.044 | 0.043 | 0.033 | 0.040 | 0.042 | 0.040 | 0.029 | 0.033 | 0.031 | 0.030 | 0.026 | |
| **19** V. jobiensis | 0.127 | 0.119 | 0.116 | 0.111 | 0.094 | 0.090 | 0.096 | 0.096 | 0.096 | 0.105 | 0.098 | 0.101 | 0.098 | 0.104 | 0.096 | 0.094 | 0.101 | 0.103 |
| **16S rDNA** | | | | | | | | | | | | | | | | | | |
| **1** Varanus sp. Louisiades | | | | | | | | | | | | | | | | | | |
| **2** Varanus sp. Admiralties | 0.010 | | | | | | | | | | | | | | | | | |
| **3** V. douarrha | 0.007 | 0.003 | | | | | | | | | | | | | | | | |
| **4** Varanus sp. Shortland I. | 0.010 | 0.006 | 0.003 | | | | | | | | | | | | | | | |

(*Continued.*)

**Table 6.** (Continued.)

| | | 1 | 2 | 3 | 4 | 5 | 6 | 7 | 8 | 9 | 10 | 11 | 12 | 13 | 14 | 15 | 16 | 17 | 18 |
|---|---|---|---|---|---|---|---|---|---|---|---|---|---|---|---|---|---|---|---|
| 5 | *V. indicus* | 0.011 | 0.007 | 0.004 | 0.007 | | | | | | | | | | | | | | |
| 6 | *V. tsukamotoi* | 0.010 | 0.006 | 0.003 | 0.006 | 0.007 | | | | | | | | | | | | | |
| 7 | *V. bennetti* **sp. nov.** | 0.012 | 0.009 | 0.006 | 0.009 | 0.010 | 0.003 | | | | | | | | | | | | |
| 8 | *V. lirungensis* | 0.010 | 0.006 | 0.003 | 0.006 | 0.007 | 0.000 | 0.003 | | | | | | | | | | | |
| 9 | *V. doreanus* | 0.020 | 0.010 | 0.013 | 0.016 | 0.016 | 0.016 | 0.019 | 0.016 | | | | | | | | | | |
| 10 | *V. rainerguentheri* | 0.010 | 0.006 | 0.003 | 0.006 | 0.007 | 0.000 | 0.003 | 0.000 | 0.016 | | | | | | | | | |
| 11 | *V. melinus* | 0.012 | 0.009 | 0.006 | 0.009 | 0.010 | 0.009 | 0.011 | 0.009 | 0.019 | 0.009 | | | | | | | | |
| 12 | *V. cerambonensis* | 0.012 | 0.009 | 0.006 | 0.009 | 0.010 | 0.009 | 0.011 | 0.009 | 0.019 | 0.009 | 0.006 | | | | | | | |
| 13 | *Varanus* sp. New Georgia | 0.012 | 0.009 | 0.006 | 0.009 | 0.010 | 0.009 | 0.011 | 0.009 | 0.019 | 0.009 | 0.011 | 0.011 | | | | | | |
| 14 | *V. finschi* | 0.012 | 0.009 | 0.006 | 0.009 | 0.010 | 0.009 | 0.011 | 0.009 | 0.016 | 0.009 | 0.011 | 0.011 | 0.011 | | | | | |
| 15 | *V. yuwonoi* | 0.012 | 0.009 | 0.006 | 0.009 | 0.010 | 0.009 | 0.011 | 0.009 | 0.016 | 0.009 | 0.011 | 0.011 | 0.011 | 0.000 | | | | |
| 16 | *Varanus* sp. Tanimbar | 0.013 | 0.011 | 0.009 | 0.011 | 0.013 | 0.011 | 0.014 | 0.011 | 0.021 | 0.011 | 0.009 | 0.009 | 0.014 | 0.014 | 0.014 | | | |
| 17 | *V. jobiensis* | 0.021 | 0.011 | 0.014 | 0.017 | 0.018 | 0.017 | 0.020 | 0.017 | 0.014 | 0.017 | 0.020 | 0.020 | 0.017 | 0.014 | 0.014 | 0.023 | | |
| 18 | *V. semotus* | 0.021 | 0.017 | 0.014 | 0.017 | 0.018 | 0.017 | 0.020 | 0.017 | 0.024 | 0.017 | 0.020 | 0.020 | 0.020 | 0.009 | 0.009 | 0.023 | 0.023 | |

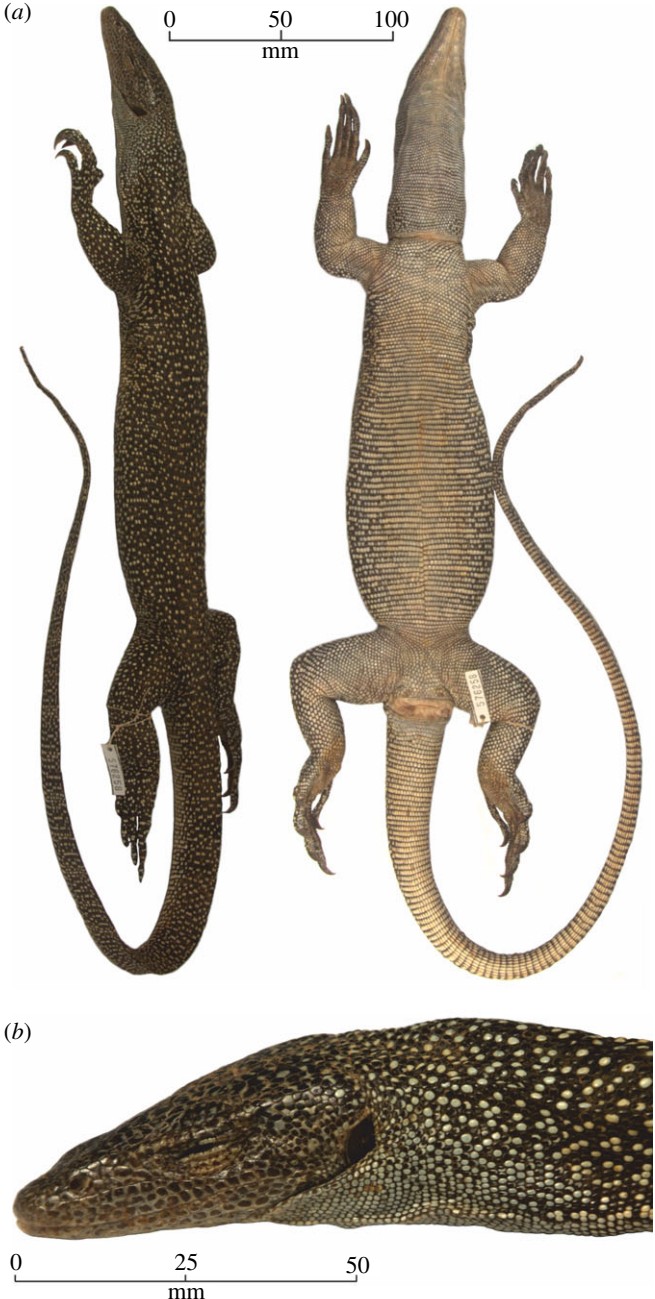

**Figure 5.** (*a*) Dorsolateral and ventral aspects of the neotype (USNM 576258) of *Varanus tsukamotoi*. (*b*) Lateral profile of the head of the neotype (USNM 576258) of *Varanus tsukamotoi*.

midbody scale counts (S: 101–126 versus 129–153 in *V. douarrha*); from *V. indicus* by its lower average scale counts around head and tail base (P: 31–40 (34.8) versus 36–47 (42) in *V. indicus*, Q: 54–74 (67) versus 60–99 (78.4) in *V. indicus*, S: 101–126 (113.5) versus 100–145 (124) in *V. indicus*) and by having dark pigmentation on the throat (versus usually cream-coloured in *V. indicus*); from *V. lirungensis* by the absence of dorsal cross-bands composed of yellow scales, dark pigmentation on the gular region (versus pink in *V. lirungensis*) and by lower scale counts around head, tail base and midbody (P: 31–40 versus 38–47 in *V. lirungensis*, Q: 54–74 versus 79–88 in *V. lirungensis*, S: 101–126 versus 134–151 in *V. lirungensis*); and from *V. rainerguentheri* by the absence of a yellow temporal stripe, by the presence of dark pigmentation on the throat (versus a mostly light-coloured throat in *V. rainerguentheri*), and by lower scale counts around tail base, midbody and along the venter (Q: 54–74 versus 76–82 in *V. rainerguentheri*, S: 101–126 versus 120–139 in *V. rainerguentheri*, T: 78–88 versus 90–94 in *V. rainerguentheri*).

### 3.4.5. Description of the neotype

A subadult specimen of undetermined sex with total length of 800 mm (SVL: 315 mm, F: 485 mm). Specimen well preserved, without signs of degradation or loss of keratin layer. Tail 1.54 times SVL, 33.5 times length of midlength height, round at base, but at 40 mm posterior to cloaca becoming increasingly compressed laterally and gaining a double scale ridge dorsally. Dorsal ground colour of trunk and limbs, head and tail black. Yellow scales—scattered singly on the dorsum and in rows of up to seven on the flanks—decorate the dorsum and dorsal sides of limbs and neck. Distal half of tail more densely spotted with yellow scales than proximal, often accumulated in latitudinal and longitudinal rows of scales, but lacking distinct cross-bands. Ventral and subcaudal scales cream coloured or grey with dark brown upper margin; scales of upper chest and gular region a mixture of cream and grey scales, some partially covered by dark brown pigmentation, appearing grey to the naked eye. Scales of ventral surfaces of limbs cream coloured, interstitial skin pale brown. Palms and soles with dark-centred, domed traction scales. Subdigital scales dark black, more or less domed and arranged in triple rows. Eight basalmost scales of the inner side of the fourth toe enlarged. Claws dark brown, sharp and recurved.

Nostrils oval, slightly compressed vertically, pointed at the rear end, surrounded by seven and eight scales, respectively. Nasal capsules slightly expanded, with a shallow groove on the rostrum. Tongue and tines dark blue-grey on both dorsal and ventral surfaces. Teeth pointed and recurved.

Dorsal head scales dark brown, polygonal and irregular in shape and size; most partly pigmented in yellow, often along one or two of the scale margins. Supralabials pentagonal, rectangular or irregular, densely covered with pits. Four scale rows between mouth nostrils, eight scale rows between nostrils across the snout. Enlarged supraoculars five on left, four on right, supralabials 23 on each side, rostral enlarged, pentagonal. Occipital damaged, scarred. Temporals irregular in size and shape, with numerous pits.

Nuchals variable in size, round to slightly oval, relatively flattened, with four to five pits, bordered by row of enlarged granules along lower and sometimes lateral margins. Smaller granules cover interstitial skin. Dorsal scales slightly more domed than nuchals or with a blunt keel, most with a single pit at posterior end. Caudal scales small, elongate rectangular, with small median ridge and single pit.

Gulars cream or grey, enlarged, rectangular or irregular near snout, quickly decreasing in size posteriorly; small and square near eye; increasing in size towards gular fold; oval black anterior border along gular fold. Many gular scales with centre dark brown. Chest scales polygonal, irregular, cream or grey, some with black anterior margin. Chest scales square with rounded posterior corners, at ribcage becoming elongate, rectangular, with black along anterior margin and row of granules along posterior margin. Ventrals with single pit at the posterior end. Subcaudals rectangular, increasingly elongate and ridged posteriorly. On distal two-thirds of tail subcaudals cream with black anterior of as much as third of scale.

### 3.4.6. Scalation

S 111, XY 129, DOR 138, T 84, VEN 100, X 39, m 86, P 35, Q 63 and R 51.

### 3.4.7. Measurements

SVL 315 mm, F 485 mm, TL 800 mm, A 47.5 mm, B 26 mm, C 18 mm, G 13 mm, H 10.5 mm.

### 3.4.8. Molecular evidence

*Varanus tsukamotoi* is resolved as a well-supported lineage (JF 94, BS 93, syn 4) in both parsimony and likelihood-based phylogenies. Its closest evolutionary relatives are the *Varanus* species to be described below (1.5/0.3% difference in ND4/16S) and *V. lirungensis* (1.4/0.3% difference), but the evolutionary affinities among these three species remain unresolved.

### 3.4.9. Variation and coloration in life

The specimens examined ($n = 43$) were generally fairly uniform in appearance. However, specimens from Pagan in the Northern Marianas were noticeably darker in coloration than specimens from other parts of the Marianas, particularly in the gular and ventral regions. Only one specimen (USNM 494380) had a light temporal stripe. Tail banding varies from weak to non-existent. Photographs of live animals [9,14] (figures 6 and 7) show them to have a black ground colour with bright yellow spots, the parts

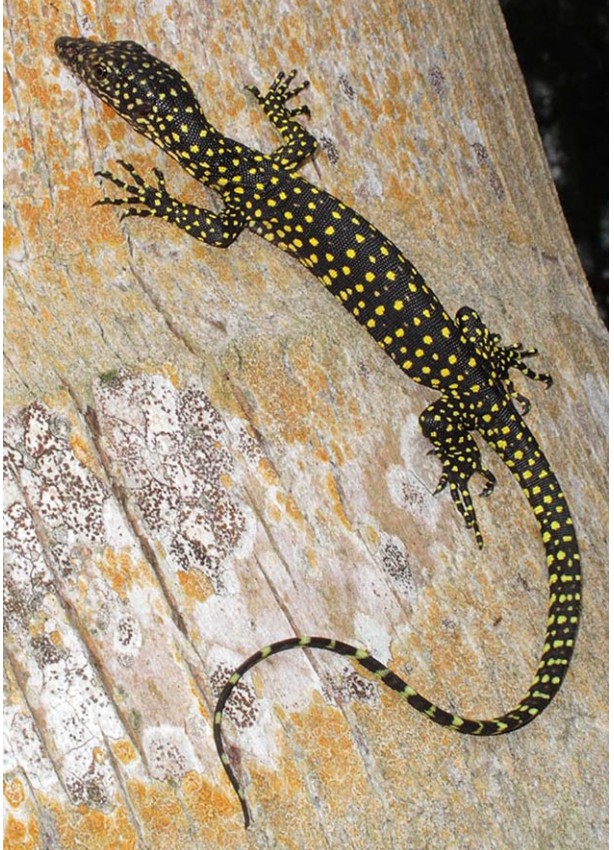

**Figure 6.** Hatchling *Varanus tsukamotoi* on Cocos Island (photo by Björn Lardner).

of the throat that are white in preservative also appear to be yellow in life. The largest specimen examined by us had a total length of 120.5 cm. Dryden & Taylor [48] reported a maximum total length of 133 cm among 100 specimens collected on Guam. A maximum weight of 2.2 kg was reported by Dryden [49] among 84 animals killed on Guam.

### 3.4.10. Etymology

Kishida named this species in honour of Dr Iwasaburo Tsukamoto, who supported his expedition to the South Sea Islands, and proposed 'Saipan monitor' and 'Tsukamoto Ohtokage' as the English and Japanese vernacular names. We suggest the common name 'Mariana monitor' as it more accurately describes the distribution of this species.

### 3.4.11. Distribution

We have examined specimens belonging to this species from Guam, Cocos Island, Saipan, Rota, Pagan, Tinian and Anatahan islands in the Mariana Islands, and from Japtan Island in the Marshall Islands (figure 1). Specimens deposited in BPBM from Aguiguan Island in the Mariana Islands were not examined by us but presumably represent this species inasmuch as the nearby islands of Guam, Rota, Saipan and Tinian all have this species. For the population on Japtan Island, a human introduction appears most likely (Fosberg [24] cited in Dryden [49]). There is a single specimen (ZMUC R4266) allegedly collected in the Bonin Islands (Ogasawara Islands) by C.B. Clausen in 1912 and examined by Philipp [50] with scale count values matching those of *V. tsukamotoi*. This record has subsequently been cited in the literature for including the Bonin Islands within the range of *V. indicus* [51,52]. However, as the Ogasawara Islands have been well surveyed by Japanese herpetologists (N. Iwai and M. Toda 2019, personal communication), including as part of gaining recent status as a UNESCO world-heritage site, and as there are no other records of *Varanus* from there, we believe this specimen represents a case of mistaken provenance.

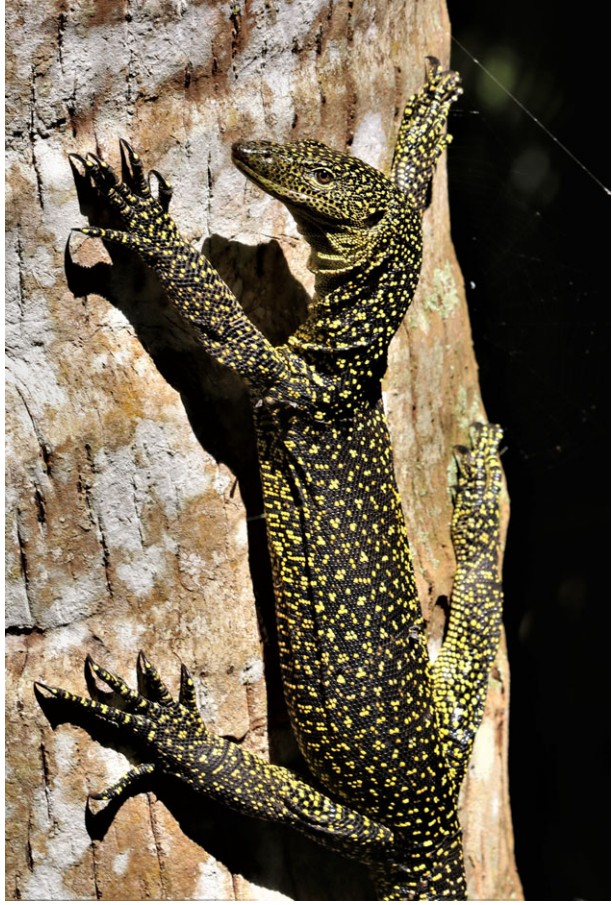

**Figure 7.** Mature *Varanus tsukamotoi* on Guam (photo by Peter Xiong).

### 3.4.12. Ecology

Dryden [49] examined the stomach contents of 84 animals dissected on Guam. The prey items found (in order of frequency) were giant African snails (*Achatina* sp.), miscellaneous arthropods (insects, insect larvae and millipedes), rats (*Rattus mindanensis* and *R. exulans*), shrews (*Suncus murinus*), hermit crabs, earthworms, slugs, bird eggs, skink (1) (*Emoia cyanurum*), gecko (1) (*Hemidactylus frenatus*), blind snake (1) (*Indotyphlops braminus*), and a skink egg. Specimens removed from Cocos Island foraged at the island's dump and frequently included chicken bones in their stomachs (FK, personal observation). Wikramanayake & Dryden [53] studied the reproductive biology of *Varanus* on Guam and considered males to be sexually mature at 320 mm and females at 275 mm SVL. Mature males averaged almost three times the mass of mature females. Reproduction appeared to be seasonal, with mating taking place during the early dry season (December–April) and eggs presumably hatching during the wet season (April–December).

### 3.5. *Varanus bennetti* sp. nov. figures 8–12

#### 3.5.1. Holotype

USNM 507504 (figure 8*a,b*), collected by Ronald Crombie, south of Ngaramasch village, Ngeaur Island, Palau, 31 July 1996.

#### 3.5.2. Paratypes

USNM 514125, 521719, Ngeaur Island, Palau. USNM 495369–70, Palau. Commonwealth of the Northern Mariana Islands: Sarigan Island (USNM 212494); Federated States of Micronesia: Losiep Island (USNM 122560), Yap Island (AMNH 00624–25, BMNH 98.5.27.1, SMF 32808–09, USNM 130186, ZMB 17520–22, 7619 and ZMH R-4727); Palau: Koror Island (AMNH 70652–53), Ngcheangel Atoll (USNM 495369).

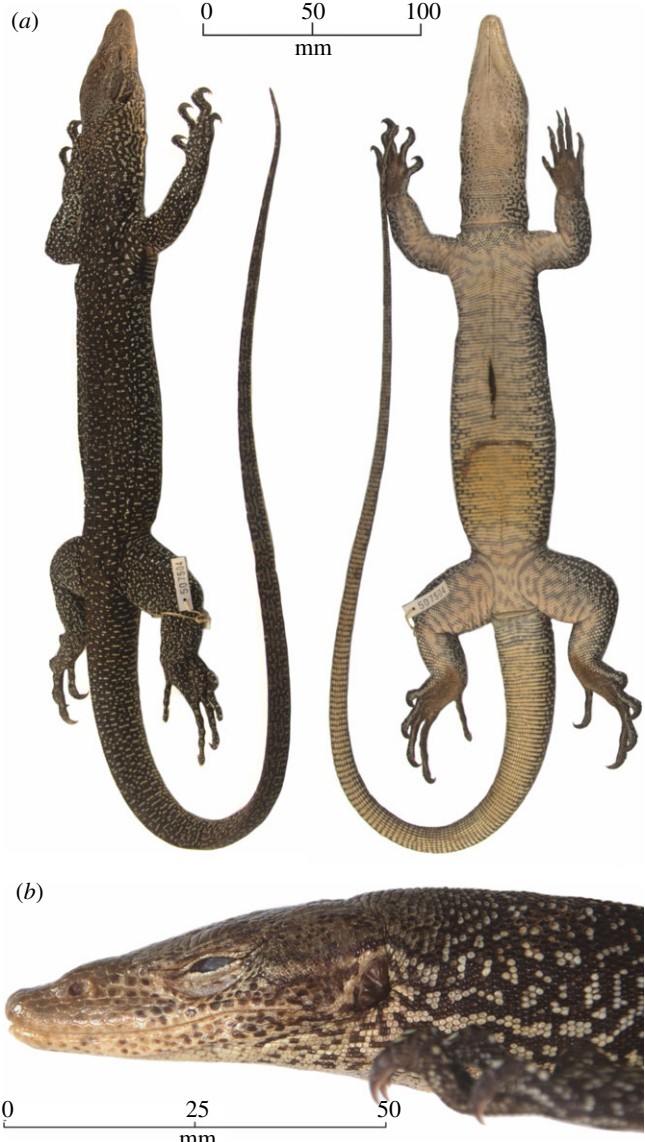

**Figure 8.** (*a*) Dorsolateral and ventral aspects of the holotype (USNM 507504) of *Varanus bennetti* sp. nov. (*b*) Lateral profile of the head of the holotype (USNM 507504) of *Varanus bennetti* sp. nov.

### 3.5.3. Diagnosis

*Varanus bennetti* can be distinguished from all other members of *Euprepiosaurus* by its unique combination of (i) dorsum black and evenly speckled with yellow scales, sometimes arranged in small groups of yellow scales, (ii) tongue dark blue/grey, (iii) venter cream coloured with pale grey cross-bands, (iv) tail exceptionally long (F/SVL mean = 1.76, range = 1.60–1.89), high XY scale counts (148–160), (v) a clear yellow temporal stripe present in about half of the studied specimens, and (vi), in life, peach colouring on the throat.

### 3.5.4. Comparisons with other members of *Euprepiosaurus*

*Varanus bennetti* can be distinguished from *V. caerulivirens*, *V. colei*, *V. doreanus*, *V. finschi*, *V. jobiensis*, *V. juxtindicus*, *V. melinus*, *V. obor*, *V. semotus* and *V. yuwonoi* by having a fully dark blue/grey tongue rather than a (at least partly) pink or yellow tongue; from *V. cerambonensis* by the absence of dorsal cross-bands and its comparatively longer tail (F/SVL: 1.60–1.89 in *V. bennetti* versus 1.32–1.61 in *V. cerambonensis*); from *V. douarrha* by the absence of dorsal ocelli and its comparatively longer tail (F/SVL = 1.60–1.89 versus 1.32–1.61); from *V. indicus* by the presence of a yellow temporal stripe in much of the population, generally higher scale counts (X: 41–48 versus 28–42 in *V. indicus*, XY:

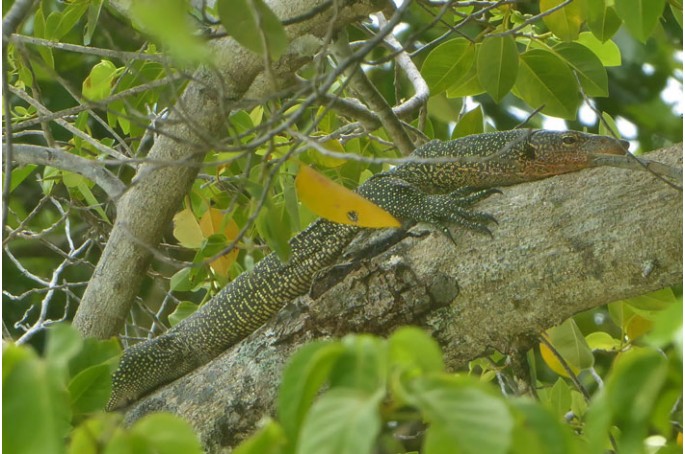

**Figure 9.** *Varanus bennetti* sp. nov., Rock Islands, Palau (photo by Thibaud Aronson).

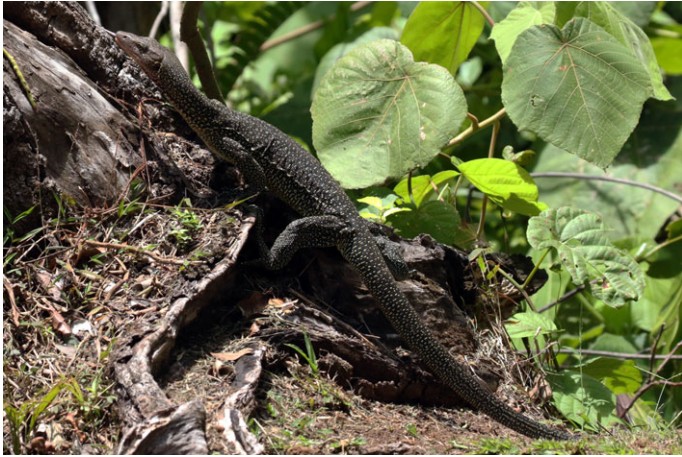

**Figure 10.** *Varanus bennetti* sp. nov., Ngarchelong, Palau (photo by Thibaud Aronson).

148–160 versus 109–158 in *V. indicus*), longer tail (F/SVL = 1.60–1.89 versus 1.22–1.70 in *V. indicus*) and dark pigmentation on the lower part of the throat (versus dark pigmentation on throat normally absent in *V. indicus*); from *V. lirungensis* by the absence of dorsal cross-bands and lower scale counts (P: 31–40 versus 38–47 in *V. lirungensis*, Q: 54–74 versus 79–88 in *V. lirungensis*, S: 101–126 versus 134–151 in *V. lirungensis*); from *V. rainerguentheri* by its longer tail (F/SVL = 1.60–1.89 versus 1.36–1.47 in *V. rainerguentheri*), higher average scale counts of all measured characters (table 4) and the peach color of the throat (versus cream in *V. rainerguentheri*); and from *V. tsukamotoi* by its longer tail (F/SVL 1.60–1.89 (1.76) versus 1.33–1.73 (1.58) in *V. tsukamotoi*), higher scale counts of all measured characters (table 4), and the peach colour of the throat of live animals (versus yellow in *V. tsukamotoi*).

### 3.5.5. Description of the holotype

Subadult specimen of undetermined sex, total length 775 mm (SVL: 275 mm, F: 500 mm). Well preserved, without degradation or loss of keratin layer. There is a 25 mm long incision on the upper abdomen. Tail muscular, long, slender (F/SVL = 1.82, 38.46 times as long as high at midlength), round at base, becoming increasingly laterally compressed and gaining a double dorsal scale ridge distal to 45 mm posterior to vent. Dorsum of trunk and limbs black with scattered yellow scales in groups of one to four. Tail black with yellow marbling, without distinctive cross-bands. Venter cream with *ca* 20 more-or-less complete grey cross-bands. Throat cream with scattered brown scales laterally and near gular fold. Head various shades of brown, with a yellow post-ocular stripe. Tongue blue dorsally and pink ventrally except for a slightly darker median line. Teeth pointed and recurved.

Nuchal scales on anterior half of neck round to slightly oval, flattened or slightly domed, bordered by row of enlarged granules along lower and sometimes lateral margins. Interstitial skin covered by smaller

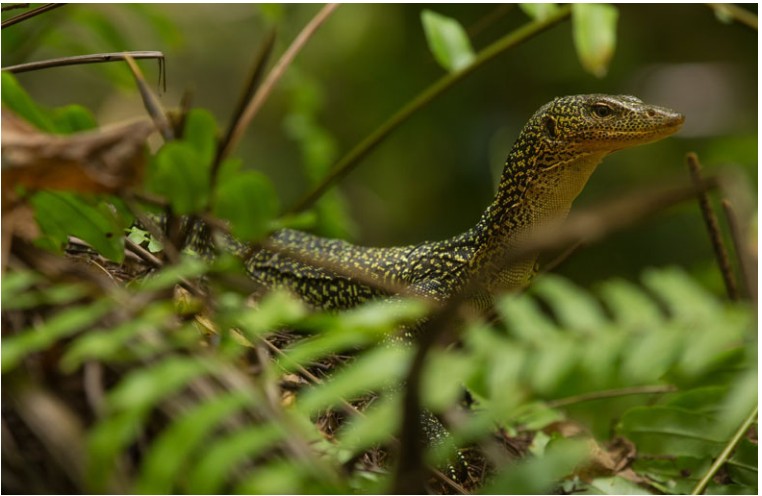

**Figure 11.** Subadult *Varanus bennetti* sp. nov., Losiep Island, Federated States of Micronesia (photo by James Reardon).

granules. Nuchal scales on posterior half of neck elongate, keeled. Dorsal scales oval to elongate, keeled, becoming rectangular and increasingly elongate distal to base of tail. Suprabrachials and antebrachials elongate, keeled, surrounded by one or more rows of granules. Suprafemorals and tibials small, elongate, keeled, surrounded by numerous small granules. Subbrachials, antebrachials, femorals and tibials polished, round to slightly oval, with row of larger granules along posterior margin.

Infracarpals and infratarsals round, domed, dark brown in centre. Claws dark brown, sharp, recurved. Gulars yellow, enlarged, rectangular or irregular toward the snout, quickly decreasing in size ventrally to level of eyes. Towards the gular fold gulars round or slightly oval, bordered by row of granules along posterior and lateral margins, with one to three darker pits. Chest scales cream or grey, irregular in shape. Ventrals cream or grey, rectangular, with rounded posterior corners and small central keel. Subcaudals cream or yellow with brown anterior margin, rectangular, elongate, with sharp central keel. Lateral caudal scales half as long as subcaudals, elongate rectangular, with central keel and pit at posterior end.

Occipital scale roundish. Enlarged supraocular scales seven on each side. Four scale rows separate mouth and naris; nine dorsal scales between the nares. Enlarged supralabials 24 on each side. Rostral pentagonal; temporals small and irregular.

### 3.5.6. Scalation

S 138, XY 158, DOR 173, T 100, VEN 118, X 47, m 109, P 45, Q 90 and R 64.

### 3.5.7. Measurements

SVL 275 mm, F 500 mm, TL 775 mm; A 44.5 mm, B 25.5 mm, C 17 mm, G 13 mm, H 11.5 mm.

### 3.5.8. Molecular evidence

*Varanus bennetti* sp. nov. is resolved as a well-supported monophyletic lineage (JF 99, BS 99, syn 6) in both parsimony and likelihood-based phylogenies. The closest evolutionary relatives of the species are *V. tsukamotoi* (1.5/0.3% difference in ND4/16S) and *V. lirungensis* (1.4/0.3% difference), but the evolutionary affinities among these three species remain unresolved (figures 3 and 4).

### 3.5.9. Variation and coloration in life

The light (probably cream coloured to yellow in life) temporal stripe is apparent in only part of the examined material. Some specimens lack tail bands altogether, whereas others show discernible bands. Crombie and Pregill [3] noted that live animals are black with prominent yellow dorsal rosettes and other irregular markings, and they have a vivid peach-coloured throat (figures 9–12, and fig. 188 in [5]). They also noted the large size (up to 180 cm) attained by *V. bennetti* in Palau, which is

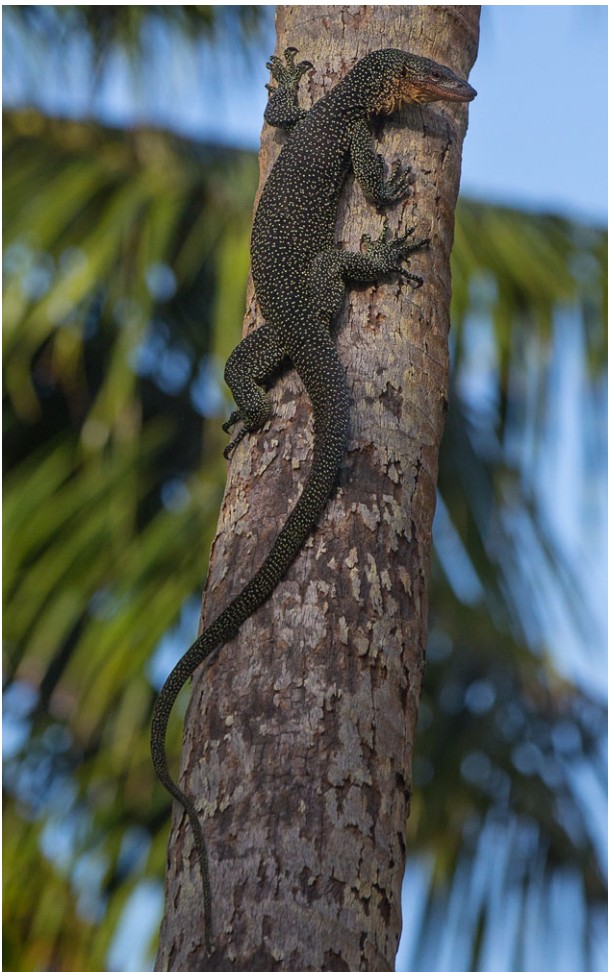

**Figure 12.** Adult *Varanus bennetti* sp. nov., Losiep Island, Federated States of Micronesia (photo by James Reardon).

exceptional within the subgenus *Euprepiosaurus*. Fourteen specimens measured in the field on Sarigan had total lengths of 67–142 cm, tail/SVL of 1.69–1.88 and weights of 200–2180 g [54].

### 3.5.10. Etymology

The specific epithet is a genitive singular patronym in commemoration of the late Dr Daniel Bennett, 1966–2020, and his life-long commitment to the study and conservation of monitor lizards in Africa and Southeast Asia. As a vernacular name we suggest 'Bennett's long-tailed monitor'.

### 3.5.11. Distribution

We have examined specimens of *V. bennetti* from Koror, Ngeaur and Ngcheangel islands in the Palau archipelago, from Yap and Losiep islands in the Federated States of Micronesia (FSM), and from Sarigan Island in the Commonwealth of the Northern Mariana Islands (CNMI) (figure 1). Crombie & Pregill [3] also list this species (as *Varanus* cf. *indicus*) from an additional two islands in the Palau group: Ngeriungs and Babeldaob.

### 3.5.12. Ecology

Crombie & Pregill [3] remarked that the monitors on Palau are decidedly terrestrial and prefer, when possible, to take refuge in terrestrial refuges rather than in trees. On Ngeaur, they are reportedly most common in the rugged limestone interior of the island [3]. Both features are atypical for species in the *V. indicus* Group, which usually seek refuge in trees and attain their highest densities in coastal habitats [55]. On Sarigan Island, the most common food items found in dissected lizards (*n* = 16) were

rats (*Rattus exulans*), insects and lizards [54]. In addition, that author found a high proportion of males among the specimens examined by him: only four of 16 specimens were females.

# 4. Discussion

## 4.1. Species recognition

The *Varanus indicus* Group *sensu* Weijola *et al*. [23] now includes 11 recognized species, and several more await description. Of these, only *V. indicus* has a wide distribution, covering most of coastal and lowland New Guinea, most of its satellite islands (including the Raja Ampat Islands, Biak, Japen, Aru, Karkar, Tolokiwa, Umboi, Sakar, New Britain, Watom, Duke of York Islands and D'Entrecasteaux Islands), and parts of the northern rim of Australia. All other species in this group are endemic to single islands or smaller archipelagos in the Moluccas, Melanesia and—as demonstrated here—in Palau, the Western Carolines and the Mariana Islands. Molecular-dating techniques have shown that expansion and diversification of this clade happened during the Pleistocene and that differentiation and speciation has been rapid on islands isolated enough for gene flow to be minimal or non-existent [23].

It is notable that there exists significant mitochondrial haplotype diversification within *V. indicus sensu stricto*. Some of these haplotypes within *V. indicus* are indeed more different to each other (by pairwise distance) than are the distances between some of the recognized species within the *V. indicus* species group. However, as highly similar mitochondrial haplotypes can often be found in geographically distant populations (e.g. the islands of Karkar, Wewak and Fergusson (figures 3 and 4), and single locations often have a mix of different haplotypes (figures 3 and 4 [23]), it seems likely there has been incomplete lineage sorting among geographically distant but recently isolated populations of *V. indicus*. In contrast, between the insular populations currently recognized as species (i.e. *Varanus bennetti* sp. nov., *V. cerambonensis*, *V. douarrha*, *V. lirungensis*, *V. tsukamotoi*), there is no evidence that gene flow or incomplete lineage sorting exists. In these instances, allopatric populations have been isolated long enough for consistent morphological, genetic and (in some cases) ecological differences to evolve, and their recognition as species is in accordance with species concepts applicable to insular populations, including both the evolutionary and unified species concepts [45,56].

## 4.2. Biogeography

Molecular dating suggests that monitor lizards reached Palau and the Mariana Islands sometime during the late Pleistocene [23]. At that time, the islands under consideration effectively occupied their present positions. The Palau, Mariana and Yap islands all lie on the Philippine Sea Plate and are the ancient remnants of intra-oceanic volcanic-arc systems that have never been in contact with any continental landmass nor with each other [57–59]. The remaining Caroline Islands east of Yap lie on the Pacific Plate [60], and their high-elevation islands are thought to derive from a former hotspot [59]. This geological history makes clear that the islands occupied by *V. bennetti* sp. nov. and *V. tsukamotoi* were all colonized either by trans-marine dispersal or by human introduction.

The close phylogenetic affiliation of *V. bennetti* sp. nov. and *V. tsukamotoi* to *V. lirungensis* from Talaud and the basal position of the northern Moluccan *V. rainerguentheri* to these three species (figure 4) suggest that the initial colonizers moving into Micronesia arrived by long-distance dispersal from the Halmahera region to Talaud, Palau, Yap and the Mariana Islands. The sequence of arrival to these archipelagos is not resolved by our dataset, but the fact that *V. bennetti* sp. nov., *V. lirungensis* and *V. tsukamotoi* form a polytomy with almost equal pairwise distances between them suggests that colonization of the three archipelagos happened in quick succession. What is striking is how these occupied islands form a straight line from Talaud in northern Central Indonesia to the Marianas. In the region of Halmahera and Talaud, the westerly South Equatorial Current (SEC) meets the barrier formed by the islands of Southeast Asia. This barrier reflects the SEC northeastward to become the North Equatorial Countercurrent (NECC), which runs directly from the Halmahera/Talaud region toward Palau, where it then continues eastward (figs 1 and 14 in [61] for illustrations). The normal reflective movement of the NECC is sufficient to explain successful rafting of ancestral *Varanus* lizards from Halmahera to Talaud to Palau. Furthermore, the NECC can vary its northward position and intensity between months [62] and years [61], making some time periods more conducive to successful transport of terrestrial biota into Micronesia. All that is needed for dispersing *Varanus* to further reach Yap or the Marianas is to ride the NECC farther eastward into the Pacific, there to be picked up by the

westward-flowing North Equatorial Current, which could deposit lizards onto the more remote islands of Micronesia along the southwestern margins of the North Pacific Gyre. We view transport along these ocean currents to be the most sensible explanation for the arrival of *Varanus* lizards to these remote Micronesian islands

The occurrence of *V. bennetti* sp. nov. on Sarigan Island in the Northern Marianas suggests that this long-distance transport mechanism has operated at least twice to deliver monitor lizards into the Mariana Islands. Theoretically, the presence of *V. bennetti* sp. nov. on Sarigan could represent either a relatively recent long-distance dispersal or a human-aided introduction. The latter explanation raises the obvious question why someone would transport lizards from Palau or the Western Carolines all the way to Sarigan—a distance of 1100–1600 km—when *V. tsukamotoi* occurs throughout the other Mariana islands (including the adjacent island Anathan, which lies merely 40 km away). Furthermore, the genetic differences seen between our Sarigan and Palauan samples of *V. bennetti* sp. nov. (figures 3 and 4) suggest the likelihood that the two populations have been separated for a period longer than humans have been in Micronesia.

Micronesian *Varanus* have long been thought to be introduced by humans throughout the region, whether by ancient Micronesians (e.g. [9]) or more recently by the Germans or Japanese during their colonial administrations of the region (e.g. [10–12,24]). Part of the reason for this interpretation has been reports following WWII of Japanese introductions throughout the region in the early twentieth century (e.g. [5]), and part of the reason was initial inability to locate *Varanus* bones in prehistoric strata in the Marianas [63]. However, later work [15] verified the species as present on Guam for at least 1600 years. Furthermore, prehistoric *Varanus* fossils tend to be rare elsewhere, even on islands where they are native or have been in the past [64,65], so the absence or rarity of fossils cannot be viewed as definitive evidence of absence of these lizards. A more sceptical view of the role of human introduction of *Varanus* in Micronesia was taken by Crombie & Pregill [3]—who argued that Palauan and Marianan populations were probably not the same species and that the former may be native— and by Cota [14], who thought that Marianan populations were native or possibly introduced by Chamorros prior to colonization by Western and Japanese powers.

Our research makes clear that at the very least several Micronesian populations of *Varanus* are native, indeed endemic, to portions of this region. This includes populations of *V. tsukamotoi* in the Marianas and populations of *V. bennetti* sp. nov. in Palau and (apparently) Sarigan. The early observations of *V. bennetti* sp. nov. in Yap [6] suggest that population is also probably native. Given that there is no evidence that native Micronesian humans used *Varanus* for food or any other purpose (e.g. [14]), there is little reason to think that those cultures transported *Varanus* around the Pacific to new islands. This, plus the great dispersal abilities of lizards of the *V. indicus* Group [23], suggest that once these archipelagos were initially colonized, further expansion to nearby islands would be relatively unproblematic. Nonetheless, it would be useful to test this idea with denser sampling within archipelagos. Our samples from Palau come from opposite ends of the main archipelago there, so it seems possible that greater genetic diversity will not be found there, although no data are currently available to test this. As noted earlier, the unique but only slightly differentiated genotype of *V. bennetti* sp. nov. on Sarigan argues for a natural dispersal event to that location. Given that Yap and Ulithi are in a straight line connecting these two extremes of the *V. bennetti* sp. nov. distribution, it seems likely that they are native too, although it remains possible that the Ulithi population is a recent introduction, as is believed by local inhabitants (J. Reardon, personal communication). What is far less settled is the status of the populations in the eastern Caroline Islands of FSM and the Marshall Islands. Traditional reports claim their derivation via Japanese introductions for rat-control purposes in the early twentieth century [5,10]; nonetheless, the earlier report of *Varanus* on Pohnpei [8] is inconsistent with that being the sole origin of all populations, and it may be that some of these populations are native too. These questions can only be resolved by denser sampling of *Varanus* populations throughout Micronesia, but our efforts to do so have been stymied by the CITES status of all *Varanus* species coupled with the fact that local governments often do not furnish CITES permits.

## 4.3. Herpetofaunal endemism in Palau and the Marianas

*Varanus bennetti* sp. nov. joins the list of reptiles endemic to Palau and the Western Carolines (as well as Sarigan), which currently consists of seven geckos, five skinks and four snakes [2,66]. The Mariana Islands have a much more depauperate endemic reptile fauna currently restricted to just one skink species (*Emoia slevini*) in addition to *V. tsukamotoi*. However, recent molecular phylogenetic

investigations have shown that at least within *Cryptoblepharus* skinks this number is an underestimate and that the endemic herpetofauna of the Marianas is slightly richer than previously thought [67].

## 4.4. Conservation

Because of the common historical presumption of non-native status of Micronesian *Varanus* and the fact that some populations are unwanted due to their predation on chickens and crabs [10,17], monitor populations have often been viewed by local inhabitants as dispensable pests, and some attempts to remove or control their numbers have been made. For example, Gressitt [17] documented the introduction of the cane toad (*Rhinella marina*) to Kayangel Atoll in Palau in an attempt to remove the *Varanus* there so as to reduce loss of chickens (*Varanus* are poisoned and die when they try to eat *Rhinella*). More recently, hundreds of *V. bennetti* sp. nov. have been culled from Angaur State, Palau, through a bounty in at least 2011, 2013 and 2014 (J. Miles 2020, personal communication); whether that bounty programme continues is uncertain. Furthermore, the *V. tsukamotoi* population on Cocos Island, just off the southwestern coast of Guam, has been subject to severe control measures so as to improve chances of introducing endangered Guam rails (*Gallirallus owsteni*) to that island [16] (D. Vice 2015–2020, personal communication) motivated in part by the belief that this monitor lizard is not native there [16,18]. Additional requests for control of *Varanus* on Guam itself were received by the U.S. Department of Agriculture in the 1990s [68], though it is not clear whether or how they were acted on, and similar control operations have been suggested as possibly required for conservation purposes on Aguiguan Island in the CNMI [69]. In 2005, the 14th Congress of the FSM requested Japanese assistance in eradicating *Varanus* from the island of Yap [5], apparently in the belief that the Japanese introduced them to that island, a belief contradicted by the report of Chamisso [6] noting their presence on Yap at that time. Currently, plans are underway to attempt eradication of the *Varanus* population on Losiep Island, just outside Ulithi Atoll, Caroline Islands (T. Hall 2019–2020, personal communication).

Given the evidence presented herein that *Varanus* populations in Palau and the Marianas are native and endemic species, attempts to control or exterminate their populations on islands in these regions must be viewed with concern and discouraged unless conservation of more-seriously threatened species may benefit from temporary reduction of *Varanus* populations. More problematic than populations in Palau and the Marianas is the still-unresolved status of *Varanus* populations throughout the Federated States of Micronesia and the Marshall Islands. Given the widespread belief in the FSM that monitors were introduced on several islands in the early decades of the twentieth century [5], it is likely that some—and perhaps many—of the populations there are not native. However, populations in the western FSM in particular may well be, given their geographic placement on a direct line between apparently native populations of *V. bennetti* sp. nov. in Palau and on Sarigan and given their long-time residence on at least Yap [6]. Unfortunately, tissue samples are unavailable from any of the many populations in the FSM or Marshalls, and so the issue cannot be addressed here. In any event, we recommend that the new taxa are included in the environmental conservation plans of their respective home countries, and that their Red List statuses are assessed in order to secure their long-term survival.

# Note added in proof

While the present paper was in press, an Opinion was published by the International Commission of Zoological Nomenclature [70] replacing the current neotype of *Varanus indicus* to one originating from the type locality Ambon. This action results in two nomenclatural changes which were not incorporated here but should be noted for future name use; 1) *V. cerambonensis* Philipp *et al.*, 1999, is synonymized with *V. indicus* Daudin, 1802, and 2) *V. chlorostigma* Gray, 1831, becomes the valid name for the species previously known as *V. indicus*. The name of the *V. indicus* Group remains unchanged.

Data accessibility. The morphological datasets have been uploaded as electronic supplementary information and all new sequence data have been deposited at GenBank.
Authors' contributions. V.W. and F.K. conceived the study; V.W. collected and analysed the morphological data; V.V. conducted laboratory work and phylogenetic analyses; A.S. and A.K. contributed with additional sequence- and morphological data. All authors contributed to the writing and revision of the manuscript and gave final approval for its publication.
Competing interests. We declare we have no competing interests.

Funding. Laboratory costs were covered by a grant from the Turku University Foundation to V.V. and V.W. The first author was supported by a postdoctoral scholarship from the Swedish Cultural Foundation. Recent collections in Papua New Guinea were made possible by grants from the National Geographic Society (to F.K. and V.W.) and the Oskar Öflund Foundation (to V.W.).

Acknowledgements. V.W. and F.K. are grateful to the National Research Institute and the Conservation and Environmental Protection Authority for permits to collect *Varanus* samples in PNG and to Bulisa Iova for providing assistance during fieldwork there. We thank the following collections for providing tissue samples: ABTC, AMS, BPBM, CAS, UMMZ, USNM and WAM. Thomas Ziegler and two anonymous reviewers provided valuable comments on the manuscript. We thank Tommy Hall, Noriko Iwai, Joel Miles, James Reardon, Mitsuhiko Toda and Diane Vice for sharing their unpublished data or other information with us. We thank James Reardon, Peter Xiong, Björn Lardner and Thibaud Aronson for letting us use their photographs.

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
