## [Reviewer comments · Royal Society Open Science]

Review History

RSOS-200092.R0 (Original submission)

Review form: Reviewer 1

Is the manuscript scientifically sound in its present form?

No

Are the interpretations and conclusions justified by the results?

No

Is the language acceptable?

Yes

Do you have any ethical concerns with this paper?

No

Have you any concerns about statistical analyses in this paper?

Yes

Recommendation?

Major revision is needed (please make suggestions in comments)

Comments to the Author(s)

Review

Weijola and collaborators produced a taxonomical study of monitor lizards in Micronesia. The study describes two new species based on morphological and molecular evidence. There are a lot of issues that I think the authors need to consider before moving forward with publication. Below, I detail my criticisms in the hope to help the authors to revise their manuscript.

Major comments

1. The title suggests that a major component of the paper will be the implications of the taxonomic decisions in the conservation and management of invasive species. However, the introduction of the paper does not contextualize this topic and most of the paper reads like a typical taxonomical study. The issue appears only at the end of the discussion. The authors need to decide what is the emphasis they will carry more on the paper and adjust the manuscript accordingly.
2. Seems to me that this work overlaps a lot with the author's previous work (Weijola et al 2019). I think the authors need to be more detailed and upfront on what are the novel contributions of this study and what were conclusions from the previous study.
3. I am not convinced that the sampling and analysis of the morphological data is bringing robust conclusions about the differences among species (more details below)
4. The molecular evidence shows that the newly proposed species are monophyletic. However, the clade that contains *V. bennetti*, *V. tsukamotoi* and *V. lirungensis* is also monophyletic. Therefore, one could argue that they are all one species. In my opinion, the author's analysis did not show conclusively otherwise. Also, the authors need a statement highlighting that their results are coming from a very limited genetic dataset, of only mitochondrial DNA. It is well reported that results can change dramatically with the addition of more DNA sequences.

Minor comments

Line 70-72 - This is a bit confusing to me. The authors refer to the region as a hotspot of diversity but state that the diversity level for reptiles is lower when compared to other hotspots. The justification used by the authors would be because of the smaller area and isolation of islands. This seems plausible if the authors were justifying the overall diversity and not only reptiles.

Line 73 - Perhaps the lower diversity level of reptiles in the region compared to other hotspots is because there are many more reptiles species to be described than for other groups (e.g. mammals and birds).

Line 77 - What exactly are you referring by "here" in this sentence? It was not clear to me.

Line 78 - What do you mean by 'interesting biogeographic puzzles'. This sentence is a bit vague.

Line 79 - How many species are currently recognized on the *V. indicus* group?

Line 92 - Could the fossil belong to a different group of *Varanus*? In that scenario, *Varanus indicus* could still be a recent introduction.

Line 132 - Why that subset of characters?

Line 131-135 - I must say that the sample sizes here for most species are very low for an LDA. The results of this analysis cannot be reliable.

Line 157-161 – I think a map with the locations of the samples used in this study would be very helpful.

Line 212-216 – See the comments above on the sample size and LDA. This may not matter because you have a more general problem regarding the sample size, but following the results of the LDA, it would be always good to follow up with cross-validation analysis and report classification error rates.

Line 253 – It seems to me that you need to mention here that you need to revalidate the name and better justify why you would need a redescription.

Line 252-255 – I am a bit confused here. Did you use any individuals from Saipan in the morphological analysis? How many?

Line 278-295 – Ideally you should have all these species from V. indicus group in your LDA, that way it would be much easier to identify potential overlap in the morphological characters. Also, I got a bit confused here because in lines 273-277 you compare with other species in V. indicus group and in the following section you compare with other species of Euprepiosaurus. From my reading (Line 79) I was assuming these were the same thing. How are they different? What are exactly the species that belong to the V. indicus group?

Line 389-411 – The same comments as above applies to this section.

Line 477 – The term 'species delimitation' here may be misleading. This term refers to a more specific set of protocols and analysis that was not performed by the authors.

Line 502-510 – What are the contributions of your study regarding this discussion?

Review form: Reviewer 2 (Robert Fisher)

Is the manuscript scientifically sound in its present form?

Yes

Are the interpretations and conclusions justified by the results?

Yes

Is the language acceptable?

Yes

Do you have any ethical concerns with this paper?

No

Have you any concerns about statistical analyses in this paper?

No

Recommendation?

Accept with minor revision (please list in comments)

Comments to the Author(s)

I found this paper to be very clear and logical. The methods were well explained as were the analysis. The figures and tables were all very helpful. It was easy to read and well written. I found the (re)descriptions of the two species to be very clear and the diagnoses to be well supported. In the early 1990's when I first saw/collected the indicus on PNG, then Palau, and

Guam it was clear they all looked different and I assumed were cryptic taxa. This is very great to finally see this work completed and done so carefully. Also, the indicus I saw on Kosrae in eastern FSM looked strikingly different from these other forms. Clearly the systematics of that set of eastern FSM populations is the next steps as outlined in their paper.

Yap is interesting and lacks several taxa such as *Cryptoblepharus* that occur in the Marianas and Palau. The *Varanus* there could easily be introduced. Yap really doesn't share any close relationship to the Palau fauna based on our unpublished genetics of other taxa there such as *Emoia atrocostata*.

Overall I found the paper to be very clean and without much need of revision.

Three small notes:

1. Line 59 spelling "revalidate"
2. References need a lot of clean-up
3. Figure 2, not sure how to interpret placement of *rainerguentheri* (ZFMK 85404) and *Varanus* sp. (AMSR 121569) Shortland as the same haplotype? That seems weird and not explanation given? This contrasts with placement on Figure 3.

Decision letter (RSOS-200092.R0)

24-Mar-2020

Dear Dr Weijola,

The editors assigned to your paper ("*Newly recognized endemism in Micronesian monitors (Reptilia: Squamata: Varanus) argues for caution in pursuing eradication plans*") have now received comments from reviewers. We would like you to revise your paper in accordance with the referee and Associate Editor suggestions which can be found below (not including confidential reports to the Editor). Please note this decision does not guarantee eventual acceptance.

Please submit a copy of your revised paper before 16-Apr-2020. Please note that the revision deadline will expire at 00.00am on this date. If we do not hear from you within this time then it will be assumed that the paper has been withdrawn. In exceptional circumstances, extensions may be possible if agreed with the Editorial Office in advance. We do not allow multiple rounds of revision so we urge you to make every effort to fully address all of the comments at this stage. If deemed necessary by the Editors, your manuscript will be sent back to one or more of the original reviewers for assessment. If the original reviewers are not available, we may invite new reviewers.

- Data accessibility

If you wish to submit your supporting data or code to Dryad (<http://datadryad.org/>), or modify your current submission to dryad, please use the following link:
<http://datadryad.org/submit?journalID=RSOS&manu=RSOS-200092>

- Competing interests

- Authors' contributions

- Acknowledgements

- Funding statement

on behalf of Dr Punidan Jeyasingh (Associate Editor) and Kevin Padian (Subject Editor)
openscience@royalsociety.org

Associate Editor's comments (Dr Punidan Jeyasingh):

I thank the two expert reviewers for their constructive comments, and invite the authors to address/incorporate these comments in a revision.

Reviewers' Comments to Author:

Reviewer: 1
Comments to the Author(s)

Review

Weijola and collaborators produced a taxonomical study of monitor lizards in Micronesia. The study describes two new species based on morphological and molecular evidence. There are a lot of issues that I think the authors need to consider before moving forward with publication. Below, I detail my criticisms in the hope to help the authors to revise their manuscript.

Major comments

1. The title suggests that a major component of the paper will be the implications of the taxonomic decisions in the conservation and management of invasive species. However, the introduction of the paper does not contextualize this topic and most of the paper reads like a typical taxonomical study. The issue appears only at the end of the discussion. The authors need to decide what is the emphasis they will carry more on the paper and adjust the manuscript accordingly.
2. Seems to me that this work overlaps a lot with the author's previous work (Weijola et al 2019). I think the authors need to be more detailed and upfront on what are the novel contributions of this study and what were conclusions from the previous study.
3. I am not convinced that the sampling and analysis of the morphological data is bringing robust conclusions about the differences among species (more details below)
4. The molecular evidence shows that the newly proposed species are monophyletic. However, the clade that contains *V. bennetti*, *V. tsukamotoi* and *V. lirungensis* is also monophyletic. Therefore, one could argue that they are all one species. In my opinion, the author's analysis did not show conclusively otherwise. Also, the authors need a statement highlighting that their results are coming from a very limited genetic dataset, of only mitochondrial DNA. It is well reported that results can change dramatically with the addition of more DNA sequences.

Minor comments

Line 70-72 – This is a bit confusing to me. The authors refer to the region as a hotspot of diversity but state that the diversity level for reptiles is lower when compared to other hotspots. The justification used by the authors would be because of the smaller area and isolation of islands. This seems plausible if the authors were justifying the overall diversity and not only reptiles.

Line 73 – Perhaps the lower diversity level of reptiles in the region compared to other hotspots is because there are many more reptiles species to be described than for other groups (e.g. mammals and birds).

Line 77 – What exactly are you referring by “here” in this sentence? It was not clear to me.

Line 78 – What do you mean by ‘interesting biogeographic puzzles’. This sentence is a bit vague.

Line 79 – How many species are currently recognized on the *V. indicus* group?

Line 92 – Could the fossil belong to a different group of *Varanus*? In that scenario, *Varanus indicus* could still be a recent introduction.

Line 132 – Why that subset of characters?

Line 131-135 – I must say that the sample sizes here for most species are very low for an LDA. The results of this analysis cannot be reliable.

Line 157-161 – I think a map with the locations of the samples used in this study would be very helpful.

Line 212-216 – See the comments above on the sample size and LDA. This may not matter because you have a more general problem regarding the sample size, but following the results of the LDA, it would be always good to follow up with cross-validation analysis and report classification error rates.

Line 253 – It seems to me that you need to mention here that you need to revalidate the name and better justify why you would need a redescription.

Line 252-255 – I am a bit confused here. Did you use any individuals from Saipan in the morphological analysis? How many?

Line 278-295 – Ideally you should have all these species from *V. indicus* group in your LDA, that way it would be much easier to identify potential overlap in the morphological characters. Also, I got a bit confused here because in lines 273-277 you compare with other species in *V. indicus* group and in the following section you compare with other species of *Euprepiosaurus*. From my reading (Line 79) I was assuming these were the same thing. How are they different? What are exactly the species that belong to the *V. indicus* group?

Line 389-411 – The same comments as above applies to this section.

Line 477 – The term ‘species delimitation’ here may be misleading. This term refers to a more specific set of protocols and analysis that was not performed by the authors.

Line 502-510 – What are the contributions of your study regarding this discussion?

Reviewer: 2
Comments to the Author(s)

I found this paper to be very clear and logical. The methods were well explained as were the analysis. The figures and tables were all very helpful. It was easy to read and well written. I found the (re)descriptions of the two species to be very clear and the diagnoses to be well supported. In the early 1990's when I first saw/collected the indicus on PNG, then Palau, and Guam it was clear they all looked different and I assumed were cryptic taxa. This is very great to finally see this work completed and done so carefully. Also, the indicus I saw on Kosrae in eastern FSM looked strikingly different from these other forms. Clearly the systematics of that set of eastern FSM populations is the next steps as outlined in their paper.

Yap is interesting and lacks several taxa such as *Cryptoblepharus* that occur in the Marianas and Palau. The *Varanus* there could easily be introduced. Yap really doesn't share any close relationship to the Palau fauna based on our unpublished genetics of other taxa there such as *Emoia atrocostata*.

Overall I found the paper to be very clean and without much need of revision.
Three small notes:

1. Line 59 spelling "revalidate"
2. References need a lot of clean-up
3. Figure 2, not sure how to interpret placement of *rainerguentheri* (ZFMK 85404) and *Varanus* sp. (AMSR 121569) Shortland as the same haplotype? That seems weird and not explanation given? This contrasts with placement on Figure 3.

Author's Response to Decision Letter for (RSOS-200092.R0)

See Appendix A.

Decision letter (RSOS-200092.R1)

Dear Dr Weijola,

It is a pleasure to accept your manuscript entitled "Taxonomy of Micronesian monitors (Reptilia: Squamata: *Varanus*): endemic status of new species argues for caution in pursuing eradication plans" in its current form for publication in Royal Society Open Science.

on behalf of Dr Punidan Jeyasingh (Associate Editor) and Kevin Padian (Subject Editor)
openscience@royalsociety.org

Associate Editor Comments to Author (Dr Punidan Jeyasingh):
Associate Editor

Comments to the Author:

I thank the authors for addressing reviewer comments. This version is much improved and I am happy to recommend it for publication.

Follow Royal Society Publishing on Twitter: [@RSocPublishing](https://twitter.com/RSocPublishing)

Appendix A

Reviewers' Comments to Author:

Reviewer: 1

Comments to the Author(s)

Review

Weijola and collaborators produced a taxonomical study of monitor lizards in Micronesia. The study describes two new species based on morphological and molecular evidence. There are a lot of issues that I think the authors need to consider before moving forward with publication. Below, I detail my criticisms in the hope to help the authors to revise their manuscript.

Major comments

1. The title suggests that a major component of the paper will be the implications of the taxonomic decisions in the conservation and management of invasive species. However, the introduction of the paper does not contextualize this topic and most of the paper reads like a typical taxonomical study. The issue appears only at the end of the discussion. The authors need to decide what is the emphasis they will carry more on the paper and adjust the manuscript accordingly.

Reply: We have added additional data on this topic to the Introduction that makes the context clearer (lines 99-103, 130-132), and we have altered the title to make it clearer that the conservation implications are immediately important to address for these species but that the paper must necessarily be focused on taxonomy before the conservation matters can logically be addressed.

2. Seems to me that this work overlaps a lot with the author's previous work (Weijola et al 2019). I think the authors need to be more detailed and upfront on what are the novel contributions of this study and what were conclusions from the previous study.

Reply: Yes, there is necessary overlap with the analyses and results of Weijola et al. (2019), but that work was focused purely on the molecular phylogenetics and biogeography of the whole subgenus Euprepiosaurus. In contrast, this current manuscript is a taxonomic and biogeographic revision of the subset of Varanus populations in Micronesia and can best be regarded as a taxonomic extension to that work. This ms. adds to the analyses of Weijola et al. (2019) two taxa absent from that work that are critical for understanding the evolution of the Micronesian species because they are both immediately basal to the Micronesian species. This fortifies and extends upon conclusions about evolutionary relationships and biogeography of the Micronesian species briefly mentioned in Weijola et al. (2019). Further, the present ms. includes a detailed morphological comparison of diagnostic differences among the Micronesian species and their closest relatives; such analysis was completely absent from Weijola et al. (2019). We make these points clearer in the last two paragraphs of the Introduction.

3. I am not convinced that the sampling and analysis of the morphological data is bringing robust conclusions about the differences among species (more details below).

Reply: The morphological differences between the proposed species (Varanus bennetti and V. tsukamotoi) are clear and significant, particularly in scalational characters, (see Fig 1a) tail/body proportions., and coloration differences. These mensural, meristic, and color-pattern features are—alone or in combination—diagnostic for species across the V. indicus Group. This same combination of features is diagnostic of our two species, consonant with the same conclusions derived from the molecular data, and of the same magnitude and general type as seen between most other recognized species within the V. indicus group. The sample sizes are actually quite good for Varanus species (especially for V. bennetti and V. tsukamotoi), and we have included as many specimens as was practically possible. We respond to this expressed concern in more detail below as the reviewer makes more specific comments.

4. The molecular evidence shows that the newly proposed species are monophyletic. However, the clade that contains V. bennetti, V. tsukamotoi and V. liruensis is also monophyletic. Therefore, one could argue that they are all one species. In my opinion, the author's analysis did not show conclusively otherwise. Also, the authors need a statement highlighting that their results are coming from a very limited genetic dataset, of only mitochondrial DNA. It is well reported that results can change dramatically with the addition of more DNA sequences.

Reply: The reviewer suggests that since V. bennetti, V. liruensis and V. tsukamotoi group together as a monophyletic clade, they could as well be called just one species. It is true that diagnosing species (i.e., identifying those characters and their geographic distribution that suggests evolutionarily independent lineages) is always a hypothesis and sometimes somewhat subjective. For that reason scientists should always carefully evaluate all available evidence. To make our interpretation as solid as possible we have used an integrative approach using a wide set of characters, basing our species boundaries on concordance between both molecular and morphological evidence.

But the reviewer seems especially swayed by molecular evidence. In this case, our phylogenetic results based on molecular data indicate that our specimens of each species form three separate clades in both parsimony and maximum likelihood analyses. Nodal support values for these three species are high (JF=99/BS=99 for V. bennetti; JF=97/BS=100 for V. liruensis and JF=94/BS=93 for V. tsukamotoi), and each species is supported by a number of synapomorphic characters (6, 4 and 4 respectively). These results show that the available molecular evidence supports the distinction of these three species.

Since both lines of evidence (molecular and morphological) clearly show that these three evolutionary lineages form cohesive and diagnostic groups that are different from each other, it is well justified to call them separate species. The reviewer suggests that we disregard this evidence and consider these three lineages as populations of a single species, but this approach would not follow good scientific practice nor common sense. Furthermore, there is more to species recognition than just molecules, and we would be curious as to what characters the reviewer would use to diagnose this alternative "species". It would contain nothing but a hodgepodge of morphological features, making it undiagnosable vis-a-vis the remaining members of the V. indicus Group. And it makes no sense at all to postulate that a single, rarely dispersing lineage represents a single species across ca. 1600 km of ocean containing only sparsely distributed small islands. Further, the reviewer's suggestion is that all these populations represent V. liruensis, yet the

populations we describe as new do not fit the diagnosis for that species. And cramming their morphological diversity into a revised so-called “*V. lirungensis*” would require such an expanded morphological diagnosis of *V. lirungensis* that it would also require that most or all other species in the *V. indicus* Group be subsumed in the same “species”, bringing us back to the state of taxonomic understanding of several decades ago, to wit: a single wide-ranging *V. indicus*.

Our dataset includes two mitochondrial genes with a total of 1000bp, this is stated on lines 175-176 and lines 199-202. We see no need to emphasize this again elsewhere in the ms., and we don't see that this is a limitation of our study inasmuch as taxa of the *V. indicus* Group are closely related (having diverged in only the past 1.3 MY), and nuclear genes would be unlikely to add much useful information to our study, given their slower evolutionary rates.

Minor comments

Line 70-72 – This is a bit confusing to me. The authors refer to the region as a hotspot of diversity but state that the diversity level for reptiles is lower when compared to other hotspots. The justification used by the authors would be because of the smaller area and isolation of islands. This seems plausible if the authors were justifying the overall diversity and not only reptiles.

Reply: In this paragraph we state that the region has been classified as one of the world's biodiversity hotspots by Conservation International and go on to discuss how this relates to reptiles. We find this to be a logical and relevant introduction to a manuscript describing two endemic species from this region. We suspect that the reviewer might not have noted the citation and therefore think that we ourselves have identified the region as a biodiversity hotspot.

Line 73 – Perhaps the lower diversity level of reptiles in the region compared to other hotspots is because there are many more reptiles species to be described than for other groups (e.g. mammals and birds).

Reply: Incorrect. Some of CI's hotspots have high reptile diversity; others have low reptile diversity. In this case, the Polynesia/Micronesia hotspot has low reptile diversity because of the small land areas involved, the remoteness of these islands to source regions, and the simplified habitats available on many of the islands (see lines 70-77). CI's hotspots were defined solely on the basis of endemic plant diversity and degree of human threat to habitats. Hence, whether reptile diversity is high or low in these areas had no bearing on their recognition and is a separate issue than the fact that they are recognized as hotspots.

Line 77 – What exactly are you referring by “here” in this sentence? It was not clear to me.

Reply: We refer to Palau as a continuation to the previous sentence. To avoid confusion we changed “here” to “Palau”.

Line 78 – What do you mean by ‘interesting biogeographic puzzles’. This sentence is a bit vague.

Reply: We have extended the sentence to clarify what we mean.

Line 79 – How many species are currently recognized on the *V. indicus* group?

Reply: We have now specified this in the text (line 81).

Line 92 – Could the fossil belong to a different group of *Varanus*? In that scenario, *Varanus indicus* could still be a recent introduction.

*Reply: The fossils were identified as *V. indicus* by the authors of the papers that reported on the findings, and we have no reason to doubt the validity of their identifications, especially inasmuch as the lead author of that paper is an expert on fossil lizards. Furthermore, only members of the *V. indicus* group are known from oceanic Micronesian islands. And the fossils are prehistoric but geologically recent, so if another species group is involved, where has it gone in Micronesia in the past thousand or so years? There is no trace of another *Varanus* species group until one reaches the Philippines, and those species are morphologically very distinct from the *V. indicus* Group. The reviewer's conjecture is not credible enough to warrant a modification to the ms. to include this unsupported speculation.*

Line 132 – Why that subset of characters?

Reply: Because the only other scalational characters (VEN and DOR) have broad overlap with T and XY and would provide little additional information in the analysis. X forms part of the XY scalecount.

Line 131-135 – I must say that the sample sizes here for most species are very low for an LDA. The results of this analysis cannot be reliable.

*Reply: The reviewer must be unfamiliar with these lizards, which are quite large and uncommonly collected. Furthermore, many of these islands are remote and poorly collected; consequently, samples from most of these islands are small. For Japtan, Palau, Western Carolines, Sarigan, Halmahera, and Talud we included the large majority of existing museum specimens. The only population (Marianas) for which we could have increased numbers much already had a large sample in our analyses, and there was no point to adding another couple dozen specimens to that. Furthermore, we use the LDA to discriminate *V. tsukamotoi* from the other three species to the southwest. In this case, sample sizes are indeed large and there is no overlap between *V. tsukamotoi* and the other species. If we were using the LDA to try to discriminate *V. bennetti* from *V. lirungensis* and *V. rainerguentheri*, we would agree with the reviewer that sample sizes are too small to make confident claims. But we don't; we use tail length and coloration features for this purpose. We now make this point clearer in the first paragraph of the Results section (lines 221-234).*

Line 157-161 – I think a map with the locations of the samples used in this study would be very helpful.

*Reply: The islands where *V. bennetti* and *V. tsukamotoi* (as well as *V. lirungensis* and *V. rainerguentheri*) occur (and from which our samples originate) are all shown in the original Fig. 7). We have now moved this figure to this part of the ms. and renumbered the figures accordingly. A map covering the full sampling for *Euprepiosaurus* is available in Weijola et*

al. 2019, but considering the more limited focus of this article we find it unnecessary to repeat that here.

Line 212-216 – See the comments above on the sample size and LDA. This may not matter because you have a more general problem regarding the sample size, but following the results of the LDA, it would always be good to follow up with cross-validation analysis and report classification error rates.

Reply: See response to the reviewer’s remark on Lines 131-135. Beyond that we are a little mystified by this request. LDA is a dimensionality-reduction technique for separating two or more classes in a more constrained space (in this case, a two-dimensional space). Cross-validation analysis is used to predict how a model will generalize to an independent data set. We are not conducting predictions here; we are identifying whether or not populations show morphological differences and identifying which characters provide the greatest discrimination. We have added a number for the percentage of correctly assigned individuals from the LDA analysis (line 221), and we hope that will assuage the reviewer’s concern.

Line 253 – It seems to me that you need to mention here that you need to revalidate the name and better justify why you would need a redescription.

Reply: *The original description and diagnosis of V. tsukamotoi was not detailed enough to differentiate the species against the increased number of species currently recognized and the original type specimen has been lost. We add this explanation to lines 269-272.*

Line 252-255 – I am a bit confused here. Did you use any individuals from Saipan in the morphological analysis? How many?

Reply: *Seven individuals from Saipan were included, as already stated on lines 269-270.*

Line 278-295 – Ideally you should have all these species from V. indicus group in your LDA, that way it would be much easier to identify potential overlap in the morphological characters. Also, I got a bit confused here because in lines 273-277 you compare with other species in V. indicus group and in the following section you compare with other species of Euprepiosaurus. From my reading (Line 79) I was assuming these were the same thing. How are they different? What are exactly the species that belong to the V. indicus group?

Reply: *We disagree with the reviewer’s first point. Multivariate tests are notoriously sensitive to numbers of samples (taxa) included. For example, if we do an LDA on most bird species, all penguins will cluster so closely together that the reviewer would call them all “the same species”. In short, the conclusions are tightly dependent on the overall morpho-space included in the study, and for discrimination among closely related species, one must be careful to include only those taxa liable to confusion with the target species. In this case we are very specifically interested in investigating morphological and molecular evolution in the Micronesian species and in deriving taxonomic conclusions from those analysis. The Micronesian species, along with V. lirungensis and V. rainerguentheri, form a small clade within the V. indicus Group that is also geographically separate from the other members of that group. Hence, including all members of the V. indicus Group in the LDA analysis would be (1) irrelevant, and (2) liable to loss of discriminatory ability in*

assessing the distinctiveness of these Micronesian taxa vis-a-vis their immediately closest relatives. The other species of the V. indicus Group are more distinct molecularly, geographically, and (in some cases) morphologically. So, confining the LDA to the four species we chose is indeed the best way to assess the taxonomic status of the two Micronesian species.

As to the confusion on lines 273-277 v line 79, we have changed the language of the former and added language to the latter to make this difference clearer. We have also indicated the species groups that comprise Euprepiosaurus in figures 3 & 4.

Line 389-411 – The same comments as above applies to this section.

Reply: se reply for comment above.

Line 477 – The term ‘species delimitation’ here may be misleading. This term refers to a more specific set of protocols and analysis that was not performed by the authors.

Reply: We changed this subtitle to ‘species recognition’ in order to avoid possible confusion. However, we also note that “species delimitation” has a much longer and general history of use than the particular set of inferences that sprang up in the past decade, so its application here is perfectly fine.

Line 502-510 – What are the contributions of your study regarding this discussion?

Reply: See lines 512-580. Again, the reviewer jumps the gun before reading the entire section.

Reviewer: 2

Comments to the Author(s)

I found this paper to be very clear and logical. The methods were well explained as were the analysis. The figures and tables were all very helpful. It was easy to read and well written. I found the (re)descriptions of the two species to be very clear and the diagnoses to be well supported. In the early 1990's when I first saw/collected the indicus on PNG, then Palau, and Guam it was clear they all looked different and I assumed were cryptic taxa. This is very great to finally see this work completed and done so carefully. Also, the indicus I saw on Kosrae in eastern FSM looked strikingly different from these other forms. Clearly the systematics of that set of eastern FSM populations is the next steps as outlined in their paper.

Yap is interesting and lacks several taxa such as Cryptoblepharus that occur in the Marianas and Palau. The Varanus there could easily be introduced. Yap really doesn't share any close relationship to the Palau fauna based on our unpublished genetics of other taxa there such as Emoia atrocostata.

Reply: Yes this is a distinct possibility that can hopefully be tested in the future once material from Yap is available for molecular analyses.

Overall I found the paper to be very clean and without much need of revision.
Three small notes:

1. Line 59 spelling “revalidate” *Reply: we corrected this.*
2. References need a lot of clean-up *Reply: we have done so.*
3. Figure 2, not sure how to interpret placement of *rainerguentheri* (ZFMK 85404) and *Varanus* sp. (AMSR 121569) Shortland as the same haplotype? That seems weird and not explanation given? This contrasts with placement on Figure 3.

Reply: We are happy the reviewer noticed this. We found that the fact some nodes were left without branch lengths was due to how nodes were collapsed when producing the strict consensus tree in the parsimony analysis. The lack of branch lengths led to false impression of V. rainerguentheri and Varanus sp. from Shortland Island sharing the same haplotype. We fixed the problem and edited the Figure accordingly.

Also note that while this ms. was in review we received additional information on bounties placed on endemic Palauan populations, and we add that information to the ms. (lines 101, 615-617) because of its conservation importance.